# SPOT: STRUCTURED PROMPTING WITH OBJECT-CENTRIC TOKENS FOR OPEN-WORLD SCENE GRAPHS

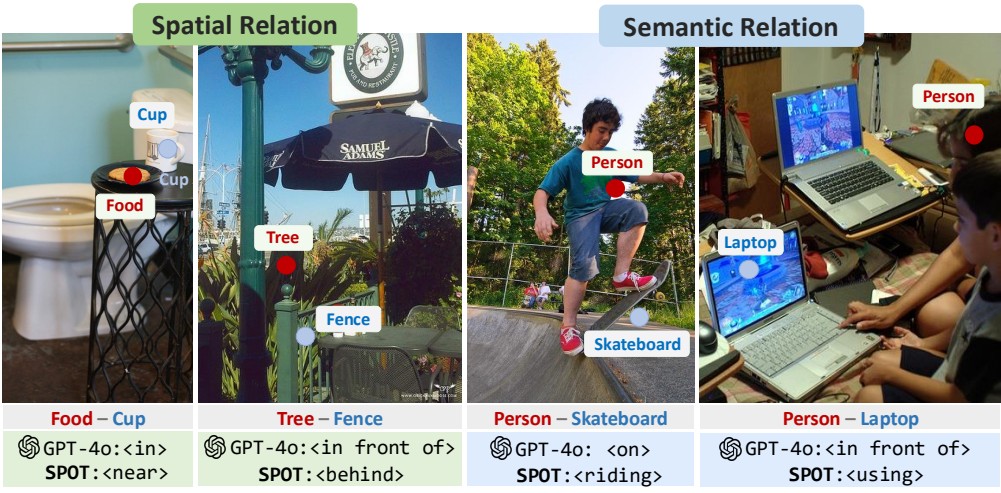

Figure 1: We propose **SPOT**, an open-source novel structured prompting framework leveraging vision language models for scene graph generation. It demonstrates superior **spatial** and **semantic** reasoning, outperforming powerful proprietary baselines like GPT-4o on open-world benchmarks.

## ABSTRACT

Scene graphs provide a compact and structured representation of visual scenes by capturing objects and their relationships, making them valuable for downstream tasks in vision-language reasoning and robotics. While early work focused on closed-vocabulary settings, newer efforts have shifted toward open-world scene graph generation (SGG) to better handle diverse real-world scenarios. Recent works explore leveraging VLMs and LLMs in open-world settings for their broad, open-vocabulary knowledge. However, existing approaches often rely on proprietary models like GPT-4o and are limited by the unstructured output behavior and weak spatial and object-level reasoning capabilities of pretrained models. We introduce SPOT, a structured prompting framework that augments open-source VLMs with spatial reasoning abilities for scene graph generation with minimal training. By combining object-centric visual features with the model's knowledge priors, SPOT achieves competitive or superior relation prediction compared to large proprietary models. Additionally, SPOT demonstrates strong cross-domain generalization, including extension to 3D scenes. Our approach is built upon open-source models, offering a scalable and accessible framework for harnessing VLMs for SGG.

## 1 INTRODUCTION

Understanding complex scenes has long been a core challenge in computer vision. Among various representations, scene graphs have emerged as a powerful paradigm to capture both semantic and spatial relationships between objects in a structured and interpretable format. By abstracting a scene into a graph with objects as nodes and their relationships as edges, scene graphs align closely with human perception and reasoning, providing a symbolic abstraction useful for downstream tasks (Johnson et al., 2015). In 2D settings, scene graphs enhance general-purpose vision-language models (VLMs) by enabling explicit relational understanding, benefiting applications such as visual question answering (Hildebrandt et al., 2020; Park et al., 2021; Liang et al., 2025). In robotics, 3D

scene graphs (Armeni et al., 2019) have gained prominence as a compact and expressive modality for high-level perception, planning, and interaction within physical environments (Chang et al., 2022; Yin et al., 2024; Werby et al., 2024). They offer a bridge between raw sensory data and the structured world, supporting effective decision-making and spatial reasoning. Consequently, Scene Graph Generation (SGG), the task of automatically constructing such graphs from sensory inputs like images or point clouds, has become a foundational problem with growing attention in both vision and robotics communities.

The development of scene graph generation began with focus on improving prediction accuracy within domain-specific, closed-vocabulary datasets (Xu et al., 2017; Yang et al., 2018; Li et al., 2018; Tang et al., 2019). While these efforts also addressed key challenges like the long-tail problem (Chen et al., 2019; Li et al., 2022a), the fixed and predefined categories limit real-world applicability. To address this limitation, the field has shifted towards open-vocabulary scene graph generation (He et al., 2022a; Yu et al., 2023; Zareian et al., 2021; Kong & Zhang, 2025; Gu et al., 2024; Li et al., 2024b; Chen et al., 2024) by leveraging open-vocabulary detectors (Liu et al., 2023; Kirillov et al., 2023) to incorporate broad semantic knowledge into the pipeline. However, they are still constrained by predefined known relations or object categories when generalizing to unseen scenarios. Most recent works propose to leverage the VLMs, to achieve a more comprehensive open-world setting (Chen et al., 2023; Li et al., 2024b). Despite these advancements, these methods often depend on closed-source proprietary models, unconstrained prompting text, and insufficient visual reasoning, frequently producing vague or physically implausible relation predictions.

In this work, we introduce **S**tructured **P**rompting with **O**bject-centric **T**okens (**SPOT**) for open-world scene graph generation. SPOT is a scalable and fully open-source framework that combines pretrained VLMs with new techniques to enhance their relation reasoning capabilities. Our approach introduces three key components to address existing challenges of using VLMs for scene graph generation. First, we design a template-based structured prompt (in contrast to the free-form prompts of prior works (Li et al., 2024b; Gu et al., 2024)) that more precisely guides the model to produce comprehensive scene graphs both out-of-the-box and after refinement through finetuning. Next, we encourage the model to consider the visual scene layout through integration of an object-centric visual feature when predicting relations. This additional signal improves upon the VLM's standard processing of the image, increasing spatial alignment and relation accuracy. Finally, to enable open-world prediction with an external object detector and no pre-defined vocabulary, we propose leveraging spatially aware pruning and integrating flexibility during evaluation to minimize penalties for semantically similar predictions. This approach goes beyond the protocol which exhaustively constructs a fully-connected graph over all object pairs, which proves computationally expensive and suffers from redundant and irrelevant relation predictions.

Overall, our framework achieves superior performance on both in-domain (e.g., Visual Genome (Krishna et al., 2017)) and cross-domain (e.g., PSG (Yang et al., 2022), 3DSSG (Wald et al., 2020a)) evaluations compared to preceding works. Beyond the strong performance on 2D benchmarks, SPOT extends naturally to 3D scene graph generation through the modular three-stage pipeline introduced in Gu et al. (2024) and outperforms recent methods and other 2D SOTA adaptations, producing more spatially accurate and semantically coherent graphs. The quantitative evaluation demonstrates the superiority, versatility, and broad applicability of our model in the physical environment.

## 2 RELATED WORKS

**Scene Graph Generation.** SGG aims to represent visual scenes as a compact graph, identifying objects as nodes and their interrelations as edges. Prior works have tackled SGG by leveraging graph convolutional networks (Yang et al., 2018), end-to-end DETR-style architectures (Li et al., 2022b), or novel strategies (Xu et al., 2017; Suhail et al., 2021; Knyazev et al., 2021). Further works target mitigating the long-tail problem through unbiased learning (Chen et al., 2019; Li et al., 2022a; Tang et al., 2020; Chiou et al., 2021), aggregating more diverse visual concept contexts (Tang et al., 2019; Zareian et al., 2020; Zellers et al., 2018), and reducing annotation cost by harnessing language-captions and leveraging weak-supervised learning (Zhang et al., 2023b; Li et al., 2022c; Zhong et al., 2021; Yao et al., 2021). However, these approaches focus on a closed-vocabulary setting, where both object and relationship classes are limited to a predefined set from datasets. While these models demonstrated strong performance on benchmark datasets, their reliance on a fixed vocabulary fundamentally limits their applicability in real-world, open-world scenarios.

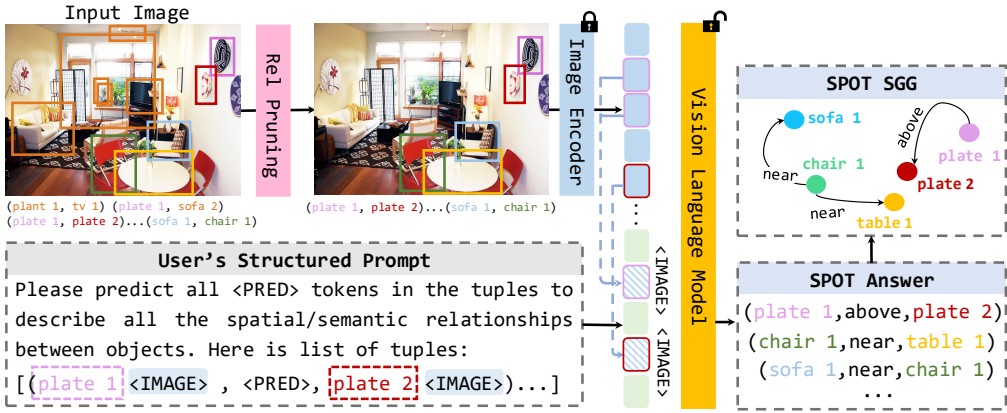

Figure 2: **SPOT framework.** Given an RGB image and object detections, our framework prunes spatially implausible relation pairs and constructs a structured prompt for scene graph generation. Our model further extracts object-centric visual feature embeddings by averaging over the corresponding image feature patches inside the bounding box areas, and injecting these into the prompt.

**Open-Vocabulary Scene Graph Generation.** Recent works have focused on extending SGG from closed-vocabulary to broader open-vocabulary settings. Most approaches approach generalizing to open-world settings by targeting either generalization to new object pairings (He et al., 2022a; Yu et al., 2023; Zareian et al., 2021; Kong & Zhang, 2025) or predicting unseen predicates between seen object entities (Zareian et al., 2021; Kong & Zhang, 2025). However, these methods are still constrained by either predefined known relations or object categories when generalizing to unseen scenarios. Another line of work jointly addresses generalization to both new objects and predicates (Zhang et al., 2023a; Chen et al., 2024), but they still train on closed-vocabulary datasets, limiting true generalization. In contrast, the recent progress in foundation models trained on large-scale datasets presents a promising avenue towards truly open-world prediction in scene graph generation pipelines.

**LLM-based Scene Graph Generation.** Leveraging LLM and VLMs for SGG has become a popular direction due to pretrained models' strong zero-shot performance on scene understanding tasks. Some works formulate SGG as a language modeling task (Kim et al., 2024; Chen et al., 2023), but require ground-truth image captions without integrating visual input for VLM query. Other works (Elskhawy et al., 2025; Liu et al., 2025) directly ask VLMs to specify the relation between one object pair at a time as an intermediate step for further relation extraction or selection, which is computationally expensive with $O(n^2)$ query times. Another direction of work uses VLMs as a direct relation prediction module (Li et al., 2024b; Park et al., 2025); however, they use either a simple prompt or rather free-form relation querying, leading to redundant or implausible output. In contrast to these, our model adapts the VLM by leveraging the structured prompt and object-centric visual features to constrain the output space with an additional relation pruning module for efficient query.

## 3 METHOD

We introduce SPOT, a structured prompting framework that augments open-source VLMs with spatial reasoning abilities for open-world scene graph generation with minimal training. Our method, illustrated in Fig. 2, addresses key limitations that arise while leveraging VLMs for this task and builds robust open-world relation prediction through three key components. First, to mitigate hallucinations and low coverage issues common in free-form generation (Gu et al., 2024), we introduce a **structured, fill-in-the-blank prompt** (Sec. 3.1). This focuses the VLM on predicting relations only for filtered, highly plausible object pairs. Second, to address the model's lack of spatial awareness (Tong et al., 2024; Yang et al., 2025b; Zhang et al., 2025; Yang et al., 2025a), we explicitly inject **object-centric visual features** (Sec. 3.2) to guide the model to harness both visual evidence and textual world knowledge priors jointly to predict object relationships. Finally, we integrate these components into an **open-world detection and pruning pipeline** (Sec. 3.3) which efficiently proposes and filters object pairs before the final relation prediction step, enabling open-world scene graph generation.

### 3.1 STRUCTURED PROMPT OVER SELECTED RELATIONS

Our first contribution is the design of a structured prompt to query the VLM to produce output relations over only the most likely relevant object-relation-object triplets. In our focus on open-world

scene graph generation, one of the challenges that emerges is the proper use of a open-world object detector which may detect many objects within even a simple scene. Often many of these objects themselves may be of little relevance for understanding a scene and moreover the relationships between unrelated and spatially disjoint objects may be irrelevant for solving a downstream task.

**Relation Proposal.** To mitigate these issues, we decouple the process of relation selection from relation prediction. Specifically, we apply a rule-based filtering mechanism to identify object pairs that are likely to interact in a meaningful way based on spatial proximity and visibility.

To construct a comprehensive scene graph, one could consider all pairwise combinations of detected objects as candidate relation triplets. However, this leads to quadratic growth in the number of pairs, introducing excessive computation and numerous semantically irrelevant or noisy relations. In particular, object pairs that are spatially distant are unlikely to share meaningful interactions and often degrade the quality of the generated scene graph.

To address this, we apply a spatial filtering strategy based on object distance, scaled by object sizes. Given the bounding boxes of two objects, which have 2D center locations, $c_i$ and $c_j$, and diagonal bounding box size, $d_i$ and $d_j$, we compute the normalized size-aware distance as:

$$d_{ij} = \sqrt{d_{geo} d_{size}} \, , \tag{1}$$

$$d_{geo} = \frac{||c_i - c_j||_2}{\sqrt{H^2 + W^2}} \qquad d_{size} = \frac{||c_i - c_j||_2}{||c_i - c_j||_2 + (d_i + d_j)/2} \, , \tag{2}$$

for an image of size $H \times W$. Once computed, we then filter pairs for which $d_{ij} > \tau$ for a fixed threshold (we use $\tau = 0.8$ in our experiments). This filtering step discards pairs of objects which are less likely to have meaningful relations while effectively reducing the number of relation queries and improves the efficiency and quality of scene graph generation.

**Structured Prompt.** Prior methods for scene graph generation with VLMs (Gu et al., 2024; Li et al., 2024b; Park et al., 2025) prompt the VLMs in a free-form manner; i.e., lists a set of objects and asks for the model to directly generate a list of relation triplets $(o_i, r_{ij}, o_j)$ between all objects from an input image. Although straightforward, this free-form prompting often results in low coverage of relation pairs in the output scene graph, with the model omitting important relation pairs while including spatially irrelevant ones.

We instead propose to prompt the VLM with a fill-in-the-blank format to predict the relation predicates for only the pairs filtered above. Our prompt follows the structure: `Please fill in the <PRED> space in the following triplets:[(obj1, <PRED>, obj2),(obj2, <PRED>, obj3),...]`, where the **<PRED>** is the placeholder for prediction in the VLM. Using this structured prompt template effectively constrains the model's output space, resolving the issues of incomplete relation pairs and degenerate scene graphs, and achieving a higher recall score with more complete graphs.

In addition to simplifying the task by constraining the output space, our structured fill-in-the-blank prompt provides the model with a structured process for reasoning about individual relation pairs rather than memorizing outputs during finetuning. This effectively suppresses hallucinations and improves overall accuracy in both in-domain and cross-domain evaluation.

To demonstrate the effectiveness of the proposed template, we evaluate four different pretrained VLMs (LLaVA-OV, Qwen2.5-VL-7B-Instruct, InternVL3-8B, and GPT-4o) without finetuning on the VG (Krishna et al., 2017) and PSG dataset (Yang et al., 2022) with free-form (Gu et al., 2024) vs. our proposed prompt template in Tab. 1. We evaluate recall at top k (R@k) predictions, and another "recall w/ sim" score is used to eliminate the vocabulary gap, which is further detailed in Sec 4.2. Our structured prompt uniformly brings significant boosts to the base models, even without any finetuning. This highlights the importance of carefully curating both the set of object-object relations requested as well as the specific format of the request.

## 3.2 OBJECT-CENTRIC VISUAL FEATURE EMBEDDING.

Language models (and VLMs by extension) are known to possess powerful priors about common world knowledge. So, it naturally follows that when asking a VLM to predict an object-object relationship, the output prediction will be heavily biased by the prior relations observed between those objects. For example, even without seeing a visual input, a VLM is biased to predict that the

Table 1: **Quantitative comparison of free-form prompting vs. our structured prompt template.** We compare the effect of using *Free-form* versus *Structured* prompts across four VLMs: LLaVA-OV-7B, Qwen2.5-VL-7B-Instruct, InternVL3-8B, and GPT-4o. Results are reported on the VG and PSG datasets, using Recall@50/100 and Recall with Similarity@50/100 as evaluation metrics.

| Model Backbone | Prompt Type | VG (1K) | | PSG (1K) | |
| --- | --- | --- | --- | --- | --- |
| | Structured Prompt | R@50/100↑ | R w/ Sim@50/100↑ | R@50/100↑ | R w/ Sim@50/100↑ |
| *LLaVA-OV-7B* (Li et al., 2024a) | – | 0.145 / 0.145 | 2.73 / 2.73 | 1.72 / 1.72 | 8.56 / 8.56 |
| | ✓ | **3.17 / 3.44** | **17.2 / 19.13** | **2.47 / 2.75** | **23.4 / 25.6** |
| *Qwen2.5-VL-7B* (Bai et al., 2025) | – | 1.40 / 1.40 | 7.88 / 7.88 | 1.17 / 1.17 | 6.97 / 6.97 |
| | ✓ | **3.71 / 4.23** | **18.6 / 21.1** | **3.37 / 3.71** | **21.6 / 24.0** |
| *InternVL3-8B* (Wang et al., 2025) | – | 6.01 / 6.01 | 13.0 / 13.0 | 5.56 / 5.56 | 12.3 / 12.3 |
| | ✓ | **8.65 / 9.45** | **23.6 / 26.4** | **11.6 / 12.8** | **27.8 / 30.8** |
| *GPT-4o* (OpenAI et al., 2024) | – | 4.55 / 4.55 | 12.1 / 12.1 | 4.70 / 4.70 | 17.7 / 17.7 |
| | ✓ | **13.5 / 15.2** | **34.8 / 39.4** | **7.74 / 8.52** | **29.7 / 34.2** |

relationship between "cup" and "table" is "on" to indicate that the cup rests on the table. While this textual bias contributes positively to overall relation prediction performance, it diminishes visual grounding. This limitation becomes critical in scenarios where spatial ground truths deviate from common expectations, for example, a cup actually being "under" a table.

To address this issue, we improve spatial grounding by embedding object-level implicit coordinate-aware visual features into the prompt. Prior works have demonstrated the effectiveness of object-level feature encoding through different mechanisms such as explicit visual annotations (Qi et al., 2025) or aggregated multi-modal features from external pretrained encoders (Huang et al., 2024). While the standard VLM input may be a set of encoded image tokens together with text tokens from the encoded prompt, we additionally add an object-specific visual token after each text object reference. Rather than relying solely on object names, we extract local visual features for each object by mapping its 2D bounding box locations back onto the spatial feature map of the pretrained vision encoder SigLIP (Zhai et al., 2023), as illustrated in Fig. 2.

Specifically, given visual encoder features $F \in R^{H \times W \times D}$ containing positional information, we identify the set of patches $P_o \subset F$ that fall within the object's bounding box, and compute the object embedding as $f_o = \frac{1}{|P_o|} \sum_{p \in P_o} p$. This average-pooled patch feature $f_o$ captures both localized visual appearance and spatial positioning. We then concatenate $f_o$ with the object's text label within the input prompt, thereby encouraging the model to attend to both semantic and spatial positioning cues. In contrast with the VLM's native global processing of the image, this direct concatenation injects the object's visual features directly alongside its textual reference in the prompt. By providing a more direct binding between vision and language for each object, we steer the model away from relying on texture priors alone and toward leveraging the visual evidence. As demonstrated in Tab. 8, this coordinate-aware visual enhancement leads to an improvement in relation prediction accuracy.

### 3.3 FRAME-WISE SCENE GRAPH GENERATION AT INFERENCE TIME

During training, we assume access to ground-truth relation triplets and corresponding 2D object coordinates, which enables direct supervision of the relation prediction module. However, at inference time, ground-truth annotations are unavailable, and the model must generate both object and relation proposals directly from raw images.

We adopt GroundingDINO (Liu et al., 2023) as our open-vocabulary object detector. The detector can be prompted with either a predefined list of object categories or a generic prompt such as "detect everything in the image," depending on the dataset or application needs. The detector outputs 2D object bounding boxes and corresponding category labels for each detected object. To suppress detection noise from repeated detections with similar object names, we apply standard and cross-category NMS on the results.

Due to the open-vocabulary nature of the task, it is necessary to detect all entities in the scene and consider potential relations between them. However, prompting a VLM with all possible object pairs is computationally demanding, so we first leverage our filtering step to reduce the set of all possible object pairings into a smaller set of plausible object pairings for which the VLM will predict relations. This filtering step is essential for reducing computation overhead for VLMs and improving the relevance of predicted relations.

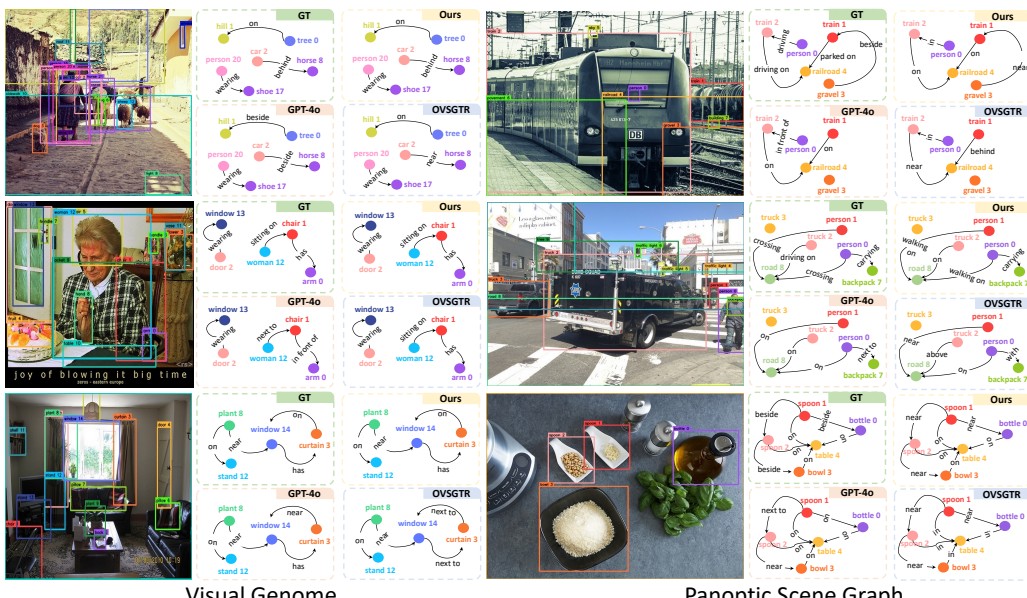

Figure 3: **Qualitative comparisons of scene graph generation.** We show the input image with detected objects and the predicted scene graphs from different methods: GPT-4o, OvSGTR, Ours, and the ground truth (GT). Our method produces more semantically accurate and spatially grounded relations, aligning closely with GT across both Visual Genome and Panoptic Scene Graph datasets.

## 4 EXPERIMENTS

### 4.1 IMPLEMENTATION

To ensure accessibility and reproducibility, we adopt LLaVA-OV-7B (Li et al., 2024a) as our baseline model and backbone for our approach. We finetune the model using a curated set of general 2D scene graph data. During training, all components except the visual encoder are unfrozen for tuning. We train the model using the Adam optimizer with a learning rate set to $1e-5$ and a batch size of 16 for 1 epoch. To support multimodal reasoning, we introduce special tokens <IMAGE> as placeholders for the averaged object-centric image visual embedding, within our structured prompts. All experiments are conducted on 8 NVIDIA A40 GPUs.

### 4.2 DATASETS AND EVALUATION METRICS

**Datasets** We train on the Visual Genome 150 (Krishna et al., 2017) (VG150) dataset, which contains 150 object categories and 50 relation types. This dataset offers a broad range of both semantic and spatial relationships. We follow the standard dataset split, using 70k images for training and 30k images for validation and testing. To evaluate the cross-domain generalization ability and application of SPOT for open-vocabulary 3D scene graph generation, we further test on the Panoptic Scene Graph (PSG) (Yang et al., 2022) and 3DSSG (Wald et al., 2020a) datasets using their test subsets.

**Evaluation Setting** We evaluate relation prediction performance on both in-domain and cross-domain settings. For the VG150 benchmark, we adopt the standard closed-vocabulary evaluation protocol and compare with existing 2D scene graph generation methods. Since the ground-truth relations in VG150 are known to be incomplete, direct accuracy metrics may unfairly penalize semantically plausible predictions. Therefore, we follow the evaluation setup in (Xu et al., 2017) and use Recall@50/100, which measures the fraction of ground-truth relations being recalled among the top 50/100 model predictions. To evaluate the performance of SPOT both as an independent relation prediction module and within an end-to-end scene graph generation pipeline, we use 2 settings: PredCLS and SGDet. In PredCLS, ground-truth object boxes and labels are provided, allowing isolated evaluation of relation prediction. In SGDet, the full pipeline is evaluated end-to-end using detected objects from GroundingDINO with our relation proposal step as input to the model. For the comparisons in the ablation study section, we report standard accuracy for the PredCLS setting on both the VG150 (in-domain) and PSG (cross-domain) test sets to explore the design space of SPOT.

Table 2: **In-domain SGG (PredCLS) results on VG150**.

| Type | SGG model | PredCLS | | |
|------|-----------|---------|---|---|
| | | R@50 | R@100 | Acc. |
| No-VLM | VCtree (Tang et al., 2018) | 50.1 | 52.5 | - |
| | GCA (Knyazev et al., 2021) | 51.2 | 53.4 | - |
| | EBM (Suhail et al., 2021) | 52.8 | 54.9 | - |
| | SVRP (He et al., 2022b) | 54.4 | 56.4 | - |
| | OvSGTR (Chen et al., 2024) | **60.5** | 61.9 | 60.8 |
| VLM | PGSG (Li et al., 2024b) | 33.8 | 40.2 | 49.2 |
| | SPOT | 58.2 | **63.4** | **70.3** |

Table 3: **In-domain SGG (SGDet) results on VG150**.

| Type | SGG model | SGDet | |
|------|-----------|-------|---|
| | | R@50/100 | mR@50/100 |
| No-VLM | VCTree (Tang et al., 2018) | 27.9 / 31.3 | 7.16 / 8.35 |
| | FCSGG (Liu et al., 2021) | 18.6 / 22.5 | 3.20 / 3.90 |
| | SVRP (He et al., 2022b) | 31.8 / 35.8 | **10.5 / 12.8** |
| | VS3 (Zhang et al., 2023a) | 34.5 / 39.2 | – |
| | OvSGTR (Chen et al., 2024) | 36.4 / 42.4 | 7.20 / 8.80 |
| VLM | PGSG (Li et al., 2024b) | 20.3 / 23.6 | **10.5** / 12.7 |
| | SPOT | **38.1 / 44.8** | 10.3 / 12.7 |

However, in a cross-domain setting, direct accuracy can be misleading due to vocabulary mismatch. For example, predicted label "near" will be penalized if the ground-truth is "beside", despite both being semantically correct. To mitigate the problem, we incorporate LLM-based semantic evaluation using Qwen-7B (Bai et al., 2025) as a similarity judge. Specifically, we query the model with prompts of the form: "[system prompt], whether '[obj1] [rel1] [obj2]' and '[obj1] [rel2] [obj2]', these two phrases refer to the same spatial/semantic relationship". We report this LLM-augmented accuracy as Acc w/ sim in our result tables to reflect a more robust and semantically-aware evaluation.

### 4.3 2D Scene Graph Generation and Comparison

**In-Domain Evaluation** Since SPOT is trained on a 2D image dataset, it can be directly applied to standard 2D scene graph generation tasks. We evaluate its performance on the Visual Genome benchmark using both PredCLS and SGDet settings with prior methods in Tab. 2 and Tab. 3, and compare it qualitatively in Fig. 3. To ensure a fair comparison, we adopt the object detector (Liu et al., 2023) that has been trained on Visual Genome, eliminating the potential distribution gap. In the PredCLS setting, ground-truth object boxes and class labels are used to isolate relation prediction performance. In the SGDet setting, the model operates end-to-end with detected objects. To reduce duplicate detections, we apply cross-category NMS with a threshold of 0.7, which removes overlapping boxes and labels referring to the same entity. To address label ambiguity between the closed-vocabulary ground truth and open vocabulary detections (e.g., a "person" may be annotated as "man", "boy", or "person"), we store all candidate labels after NMS and consider a prediction correct if any of the assigned labels match the ground truth. For relation ranking to compute the recall metrics, we compute a confidence score by combining the object detection probability and the distance between the centers of the subject and object (as Eq. 1). Since for a pair of subject and object, they appear with the same score, to adapt our setting for evaluation, for one subject-object pair, it is marked correct if it matches any ground-truth triplet involving that pair. As shown in Tab. 3, SPOT demonstrates strong in-domain performance. It outperforms prior models, particularly other VLM-based approaches like PGSG, indicating superior relation prediction capabilities, as evidenced by PredCLS R@100 and accuracy. Overall, SPOT achieves robust relation prediction accuracy highlighting the effectiveness of our object-centric structured prompting design.

**Cross-Domain Evaluation** We further evaluate the generalization ability of SPOT across datasets in Tab. 4 compared with previous methods and GPT-4o. We consider two benchmarks: PSG (Yang et al., 2022) and the single-frame (2D) version of 3DSSG (Wald et al., 2020a). Importantly, no data from these datasets is seen during our model's training. PSG shares visual and categorical similarity with VG150, while 3DSSG poses a more significant domain gap, targeting indoor scenes and including unseenrelations such as "bigger than." In order to handle vocabulary mismatches in cross-domain evaluation, we adopt 2 rules for determining if relations are equivalent: we consider predicted spatial relations correct if they contain the same spatial words as the ground truth (for instance, "standing on" and "on" would refer to the same spatial relation), and we use an LLM as a judge to calculate Acc. w/ sim as described in Sec. 4.2 to credit relations that are semantically equivalent to the ground truth. Results show that SPOT maintains strong generalization and outperforms prior models, including GPT-4o with/without our proposed structured (with qualitative results shown in Fig. 1). This result validates that our framework effectively adapts the VLM for improved scene graph relation reasoning, while leveraging the VLM's general knowledge from large-scale pretraining for generalization.

Table 4: **Cross-domain SGG results on PSG and 3DSSG datasets.** We evaluate several SGG models under the PredCLS setting. SPOT achieves generally better performance across both datasets, demonstrating strong generalization and relation prediction capabilities.

| SGG model | PSG | | | 3DSSG Per Frame | | |
|---|---|---|---|---|---|---|
| | Acc. | Acc w/ sim | PredCLS R@50/100 | Acc. | Acc w/ sim | PredCLS R@50/100 |
| Llava-OV-7B (Free-form) (Li et al., 2024a) | - | - | 1.7 / 1.7 | - | - | 5.4 / 5.4 |
| GPT-4o (Free-form) (OpenAI et al., 2024) | - | - | 4.7 / 4.7 | - | - | 18.5 / 18.5 |
| GPT-4o (Our template) (OpenAI et al., 2024) | 16.6 | 52.5 | 15.8 / 17.3 | **47.8** | **75.1** | **35.2 / 36.1** |
| OvSGTR (Chen et al., 2024) | 6.1 | 39.6 | 5.3 / 5.8 | 2.91 | 49.7 | 1.23 / 1.67 |
| **SPOT** | **23.6** | **59.5** | **15.9 / 18.0** | 39.1 | 71.2 | 26.9 / 29.0 |

Table 5: **Comparison of 3DSSG performance on the whole-scene setting**. In-domain methods (top) are trained and tested on the same domain, while cross-domain methods (bottom) are not exposed to 3DSSG data during training. For cross-domain methods, we only consider spatial alignment. [†] indicates methods we adapted for evaluation.

| Type | SGG model | 3DSSG Whole Scene | | |
|---|---|---|---|---|
| | | Acc. | Acc w/ sim | PredCLS R@50/100 |
| **In-domain** | SGPN (Wald et al., 2020b) | 71.0 | - | 58.0 / 58.5 |
| | SGFN (Wu et al., 2021) | 69.0 | - | 58.9 / 59.4 |
| | USG-Par (Wu et al., 2025) | - | - | 81.7 / 84.1 |
| **Cross-domain** | OvSGTR[†] (Chen et al., 2024) | 2.67 | 71.3 | 0.45 / 1.03 |
| | ConceptGraphs (Gu et al., 2024) | - | - | 24.2 / 31.4 |
| | **SPOT (Full)** | **47.6** | **86.1** | **29.2 / 43.8** |

## 4.4 3D Scene Graph Generation and Comparison

While our model is trained on 2D scene graph generation, we also demonstrate that SPOT can be readily integrated into existing open-world 3D scene graph generation pipelines. Specifically, we adopt the modular framework introduced in ConceptGraphs (Gu et al., 2024), which leverages large-scale 2D foundation models for object discovery and relation reasoning in 3D scenes (illustrated as in Fig. 9). Our goal is to show that SPOT can serve as a drop-in replacement for the relation prediction component in this pipeline, offering improved flexibility and accuracy with a non-proprietary model, which further demonstrates the effectiveness of our proposed framework.

We use the pipeline defined in ConceptGraphs (Gu et al., 2024) to detect the set of objects $\{o_1, o_2, ..., o_n\}$ along with their 3D locations from multi-view images. The pipeline uses 2D segmentation (Kirillov et al., 2023) and detection models to extract object masks and labels. RGB-D frames, along with camera intrinsics and poses, are used to lift 2D masks into 3D via depth projection, and detections are fused across multiple views. We then apply SPOT to propose object pairings and predict their relations. We the follow the same post-processing as ConceptGraphs to aggregate predictions across frames and eliminate redundant edges. Further details are provided in Supplementary Sec. E.

We evaluate on 3DSSG in Tab. 5. While SPOT is not specifically optimized for 3D inputs, it consistently outperforms both ConceptGraphs and OvSGTR adapted to the 3D setting. These results highlight the modularity and effectiveness of our approach when applied to 3D scene understanding tasks, even without explicit 3D-specific training.

## 4.5 Spatial and Semantic Generalization Discussion

By distilling the scene graph knowledge into the network, it's interesting to further explore how well the model generalizes on spatial and semantic components separately. Intuitively, the spatial reasoning benefits from the implicit positional embeddings introduced by object-level feature input, whereas the semantic reasoning benefits from richer relational patterns present in the training data. To investigate this, we conduct an additional quantitative evaluation on PSG by manually grouping relations into "spatial" and "semantic" subsets. For ambiguous cases in the open-world setting (For example, "walking on" includes both a spatial component ("on") and a semantic action("walking")), we place them into the semantic category. As shown in Table 6, across both subsets, SPOT demonstrates consistent improvements compared to all baselines, contributing to its overall performance gains. An

Table 6: **Comparison of PSG performance on spatial and semantic subsets**. Models are compared on manually separated subsets. The model is evaluated over relation accuracy and accuracy with similarity. The input prompt remains the same for all models.

| SGG model | Spatial | | Semantics | |
|---|---|---|---|---|
| | Acc. | Acc w/ sim | Acc. | Acc w/ sim |
| *InternVL3-8B (Wang et al., 2025)* | 22.52 | 44.91 | 11.83 | 27.87 |
| *Qwen2.5-VL-7B (Bai et al., 2025)* | 6.81 | 47.65 | 7.33 | 19.77 |
| *OvSGTR (Chen et al., 2024)* | 10.15 | 44.59 | 0.45 | 39.82 |
| **SPOT** | **39.29** | **69.07** | **12.54** | **50.19** |

Table 7: **Exploring the value of language and visual priors.** When finetuning using our structured prompt, we evaluate the model's reliance on language (object names) and visual priors.

Table 8: **Spatial Information Ablation.** Including spatial information in the text prompt improves performance with our object-centric feature, slightly outperforming text coordinates.

| Input to VLM | | VG | PSG | | Spatial Information in VLM Prompt | VG | PSG | |
|---|---|---|---|---|---|---|---|---|
| vision | object names | Acc.↑ | Acc.↑ | Acc. w/ Sim↑ | | Acc.↑ | Acc.↑ | Acc. w/ Sim↑ |
| ✓ | - | 40.5 | 20.4 | 39.4 | Base *(No spatial info)* | 45.7 | 22.3 | 57.9 |
| - | ✓ | 44.5 | 19.2 | 54.5 | *+ Coordinate as Text* | 46.2 | 23.5 | 58.9 |
| ✓ | ✓ | **45.7** | **22.3** | **57.9** | *+ Object-centric Feature* | **46.3** | **23.6** | **59.2** |

interesting observation is that between two subsets, the largest performance gains are expected from spatial relationships, which may be attributed to the training data imbalance.

## 5 ABLATION

In this section, we perform further experiments to give further insight into the design of SPOT. In Tab. 1, we demonstrated that our proposed structured prompt generally improves the performance of different VLMs compared with the free-form prompting introduced in Gu et al. (2024). We now explore different design choices for: language and visual priors, object-centric embedding, and relation pruning components of our framework.

**Language and visual priors**. To explore the VLM's utilization of text prior knowledge and visual cues, we conduct ablations of the visual and textual hints provided to the finetuned VLM in Tab. 7. In the model provided with both vision and object names (last row), the image is annotated with object bounding boxes to indicate the objects' locations and categories of interest, and the prompt includes object names in triplets. For the object-names-only setting (second row), the model is provided with only the structured prompt without any image input, such that the model predicts everything based on text priors i.e. object names. Conversely, in the vision-only setting (first row), object names are masked with token <OBJ i> both on the image and in text, forcing the model to predict the relation only based on visual clues without object name text priors. The quantitative comparison results are shown in Tab. 7. Without the object name in the prompt, we see a large performance drop, indicating that the pretrained model's language priors are essential for our task. However, removing the visual input results in a smaller drop, suggesting that the model is not fully leveraging visual and spatial cues. This result motivates our introduction of object-centric visual features to improve visual grounding.

**Object-centric Feature**. As shown in Tab. 7, the model more heavily relies on textual knowledge for predicting object relations (for instance, the prior that a cup will often be on a table), driving us to include object-centric visual embeddings to encourage the model to consider relevant visual cues. In Tab. 8 we ablate our object-centric visual embeddings, and also consider the alternative design of providing the objects' bounding box coordinates directly as text into the prompt. We observe that our object-centric feature embeddings achieve superior performance both over the VLM without spatial object inputs (Base) and embedding the representing the objects as coordinate text.

**Relation Pruning**. An alternative to our relation proposal pruning approach (Sec. 3.3) is to train the VLM to learn to prune irrelevant object pairings. We consider 2 approaches: (1) Learn to classify during training: use an additional special token <UNKNOWN> to indicate that there should not exist a relation inside this object pair (first row). (2) Leverage the normalized logits as relation probability scores: select the final output logits of the chosen relation words, normalize them along the batch

dimension, and rank by score (second row). We compare the performance on both the VG and PSG datasets under the PredCLS setting in Tab. 9. The significant failure mode of method (1) is the over-prediction of <UNKNOWN> class, which suppresses the try object relationships. While most normalized probability in method (2) is rather high, making it hard to distinguish efficiently. In contrast, our design choice to prune relation pairings prior to VLM inference is effective, achieving higher Recall, and practical, requiring less compute.

Table 9: **Ablation study of relation pruning strategies.** Our distance-aware filter is compared against two straightforward alternatives: training the model to classify an <UNKNOWN> token and ranking pairs by normalized logit scores. Ours outperforms others on Recall@k, demonstrating the effectiveness of our relation pruning.

| Model Variant | VG (In-domain 1K) | PSG (Cross-domain 1K) | |
| --- | --- | --- | --- |
| | R@50/100 ↑ | R@50/100 ↑ | R w/ Sim @50/100↑ |
| <UNKNOWN> Prediction | 33.0 / 42.7 | 13.2 / 15.8 | 32.1 / 38.3 |
| Logit Score Rank | 31.8 / 43.2 | 8.81 / 12.1 | 28.2 / 37.0 |
| Ours (Distance-aware filter) | 58.2 / 63.4 | 15.9 / 18.0 | 38.7 / 44.2 |

**Object Detection**. The external detector is a crucial component in our modular framework, as its performance quality directly influences the following relation prediction. To identify the most suitable backbone model in our pipeline, we conduct a quantitative comparison among the recent state-of-the-art detection models: VLM-based model (Qwen3-VL-8B), vision foundation model (Florence-2-large), classic real-time detector (YOLO-series), and open-vocabulary detector (GroundingDINO). These models are evaluated on a small subset of the Visual Genome dataset under an open-vocabulary setting without any parameter tuning, as shown in Table 10. For models requiring the text inputs, the model dataset vocabulary space is provided, with the assumption that we have no access to per-frame groundtruth object sets. It can be observed that while VLM-based approaches report good performance when the input vocabulary space is small, they are slow (e.g., 47s/image) for dense prediction pipelines. And it struggles with increasingly more out-of-context errors when the object space grows large. Classic real-time detectors such as YOLOv11 are extremely fast, yet achieve lower recall than GroundingDINO, which would bottleneck the overall SGG performance. Furthermore, they are also closed-set and do not support flexible text-conditioned detection. Based on the empirical verification, we select GroundingDINO as our out-of-the-box solution in open-world settings.

Table 10: **Ablation study over detector design choice.** We evaluate the object detection performance over object Recall and inference speed perspectives.

| Detector | Object recall (%) | Inference speed (s/image) |
| --- | --- | --- |
| Qwen3-VL-8B (Team, 2025) | 3.48 | 47.0 |
| Florence-2-Large (Xiao et al., 2023) | 13.6 | 0.472 |
| YOLOv11n (Khanam & Hussain, 2024) | 10.2 | 0.0451 |
| YOLOv11x (Khanam & Hussain, 2024) | 10.7 | 0.0542 |
| GroundingDINO (Liu et al., 2023) | 20.9 | 0.261 |

## 6 CONCLUSION

In this work, we presented SPOT, a structured prompting framework designed to enhance open-world scene graph generation with vision-language models. By combining template-based prompts, object-centric visual embeddings, and relation pruning, our approach addresses the limitations of free-form prompting and improves both spatial grounding and semantic coherence. Extensive experiments demonstrate that SPOT consistently outperforms prior methods across in-domain and cross-domain benchmarks, achieving competitive or superior results compared to large proprietary systems such as GPT-4o. Moreover, SPOT generalizes effectively to 3D scene graph generation, underscoring its scalability and modularity. Beyond empirical gains, our framework contributes a fully open-source, efficient, and interpretable solution for scene understanding. We believe that SPOT offers a step forward toward more accessible, reliable, and generalizable scene graph reasoning, paving the way for broader applications in vision-language reasoning, robotics, and embodied AI.

## REPRODUCIBILITY STATEMENT

We will release our code and model checkpoints upon the publication of the final version. Our whole pipeline builds on top of open-source components. To further ensure reproducibility and open-source readiness, we have included extensive implementation details in the manuscript.

- **Open-weight base model**: Our framework is built entirely on the open-source LLaVA-OV-7B backbone and utilizes public benchmarks (VG, PSG, 3DSSG).

- **Implementation details**: We have provided comprehensive implementation details, especially in the Supplementary Materials, to allow for replication. Specifically, Appendix B elaborates on data preprocessing, structured prompt, hyperparameters during training; Appendix C details the inference pipeline and relation pruning; Appendix D describes the evaluation protocol and pseudo-code.

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

# Appendix

## A IMPLEMENTATION DETAILS FOR ARCHITECTURE

### A.1 OBJECT-LEVEL FEATURE EXTRACTION

An essential aspect of our feature extraction pipeline is object-level patch feature integration, which enriches relationship queries with fine-grained visual information. We explore three types of visual encoders: the CLIP image encoder, DINOv2-pretrained features, and the original SigLIP encoder used in LLaVA-OV. As shown in Table 11, the SigLIP features yield the most significant performance improvement, likely due to their stronger image-text alignment learned during pretraining. The benefit of using SigLIP is that it serves as the original vision encoder for LLaVA-OV, which not only ensures better feature alignment but also avoids introducing an additional vision encoder. For each object in a triplet, we use its downscaled ($384 \times 384 \to 27 \times 27$) 2D bounding box to query the corresponding image patch features. If a feature patch overlaps with the object's 2D region, it is included in the computation. We then average the selected patch embeddings to form the object-level feature, which is concatenated with the corresponding text embedding.

Table 11: **Ablation study of patch visual model variants.** We evaluate the impact of different visual models on the object patch features on relation prediction accuracy.

| Vision Encoder | VG (In-domain 5K) | PSG (Cross-domain 1K) | |
|---|---|---|---|
| | Acc↑ | Acc↑ | Acc w/ Similarity↑ |
| *CLIP* (*Radford et al., 2021*) | 46.2 | 22.8 | 57.3 |
| *DINOv2* (*Oquab et al., 2023*) | 46.2 | 22.5 | 57.7 |
| *SigLIP* (*Zhai et al., 2023*) | **46.3** | **23.6** | **59.2** |

### A.2 DEPTH EXPLORATION

While our method primarily relies on 2D visual and semantic cues, we explore the possibility of incorporating depth information into the relation prompting process, motivated by recent works such as VCoder (Jain et al., 2024). Unlike approaches that use explicit 3D positional embeddings or train a specialized 3D encoder (Zhou et al., 2024), requiring large-scale supervision to align with language space, we instead adopt an efficient strategy that reuses the pretrained visual encoder already aligned with the text modality for depth, as Sec. A.1. Specifically, we estimate monocular depth maps from 2D images using Depth Anything v2 (Yang et al., 2024), and normalize the predicted depth values to the [0,1] range for consistency. For inference, the model can be seamlessly used with ground-truth depth or estimated depth. After obtaining the depth map, we feed it directly into the frozen image encoder. The object-level depth features could be extracted by pooling visual encoder patch embeddings corresponding to each object's region in the depth map, as the image embedding. They are concatenated after the <DEPTH> placeholder for each object, as shown in Fig. 4, and then fed into the backbone.

Similarly to object-centric image embedding, we also explore different intuitive alternatives for depth embedding and compare quantitatively in Tab. 12. The exploration ranges from 3D positional embedding, an additional 3D encoder, and the reuse of the visual encoder, as in our model. For the 3D positional embedding, the depth is lifted into the point cloud to extract the bounding box minimum and maximum coordinates along the z-axis, and we use a fixed sinusoidal positional encoding to embed these 3D locations. For the additional 3D encoder, we use the same method to lift the point from 2D to 3D for the whole object point cloud, and use the pretrained Uni3D (Zhou et al., 2024) encoder for embedding. An additional projection layer is trained to align the input space. As observed in Tab. 12, encoding the depth map with the VLM image encoder performs better. We hypothesize this is because the image encoder is already aligned with the LLM input space, and the depth patches are naturally aligned with the image patches since they are encoded by the same model.

Empirically, given the above analysis, we observe that the inclusion of depth features yields marginal improvement in quantitative metrics. However, from a qualitative perspective, the model produces better spatial relations as shown in Fig. 13. While the improvements brought by depth are not

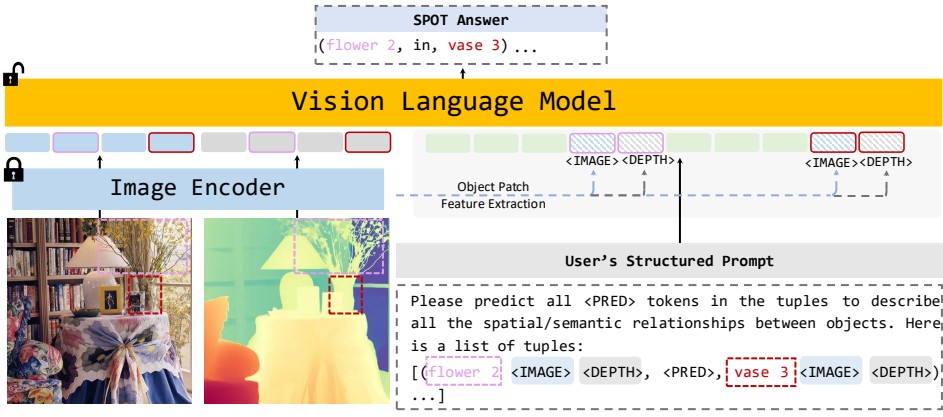

Figure 4: **SPOT method with depth for input**

Table 12: **Ablation study of different depth integration variants.**

| Depth Info Variant | VG (In-domain 5K) | PSG (Cross-domain 1K) | |
|---|---|---|---|
| | Acc↑ | Acc↑ | Acc w/ Similarity↑ |
| *3D Positional Embedding* | 46.3 | 23.2 | 58.4 |
| *3D Encoder (Zhou et al., 2024)* | 46.3 | 22.6 | 59.1 |
| *SigLIP Depth (Zhai et al., 2023)* | **46.4** | **23.6** | **59.5** |

quantitatively reflected in our current evaluation settings, we believe it presents a promising direction for future work on explicit spatial reasoning. Hence, we provide the depth integration option here for further development and investigation.

# B IMPLEMENTATION DETAILS FOR TRAINING

## B.1 DATA PREPROCESSING

In our experiments, we finetune the model on Visual Genome 150 (Krishna et al., 2017), which serves as the primary dataset to ensure fair comparison with other baseline methods presented in the main paper. Input images are preprocessed to match the expectations of the SigLIP vision encoder. All images are resized to a resolution of $384 \times 384$ and normalized using the standard mean and standard deviation values from the model's pretraining configuration. We do not apply any data augmentation techniques such as random flipping during the fine-tuning stage in order to maintain a consistent mapping between visual cues and relational language. Bounding box coordinates are extracted inside every object pair and normalized to a range of [0, 1] relative to the image dimensions before being used for object-centric feature extraction.

## B.2 STRUCTURED PROMPT DETAILS

Fig. 5 illustrates our structured prompt template with example input. Instead of sequentially querying each object pair for their relationship, we adopt a more efficient approach by constructing a structured prompt template, illustrated in Figure 5. Within this template, <IMAGE> (and <DEPTH> added alternatively) are fixed special tokens that serve as anchors for inserting object-level visual (and depth) features, as described in Section A.1. These tokens are not updated during training but are used to locate where the visual embeddings should be injected. We also introduce a special <PRED> token to indicate the position where the model should predict the relationship between the object pair. This design allows for explicit supervision and reduces redundancy in the prediction process. Additionally, each object name is followed by a numeric identifier to distinguish between different instances of the same class.

```
{
"from"  "human",
"value"  "<image>\nYou are an agent specializing in identifying the physical and
spatial relationships in images for 3D mapping.\nYour task is to analyze the
images and output a list of tuples describing the physical relationships between
objects.\nNote that you are describing the **physical relationships** between
the **objects inside** the image. \nYou will also be given a text list of
relation tuples. The list will be in the format: [("object 1", "<PRED>", "object
2"), ("object 3", "<PRED>", "object 2")].\nPlease predict all <PRED> in the
tuples.\nHere is the list of predicate tuples: [(bench 0 <IMAGE>, <PRED>, grass
6 <IMAGE>), (sky 4 <IMAGE>, <PRED>, grass 6 <IMAGE>),..., (wall 8 <IMAGE>,
<PRED>, building 7 <IMAGE>)]. Please describe the spatial relationships between
the objects in all tuples."
}
{
"from"  "gpt",
"value"  "[(bench 0, on, grass 6), (sky 4, over, grass 6), (sky 4, over,
building 7),...,(wall 8, enclosing, building 7)]"
}
```

Figure 5: SPOT structured prompt template.

## B.3 HYPERPARAMETER TUNING

During training, the vision encoder is kept frozen, while the language model and adapter layers are trainable. We do not apply AnyRes during either training or inference. The model is optimized using Adam with a learning rate of $1 \times 10^{-5}$, a cosine learning rate scheduler, and a warm-up ratio of 0.03. Training is conducted on 8 NVIDIA A40 GPUs with a batch size of 16 for 1 epoch. Due to out-of-memory issues commonly encountered with long prompt sequences, we limit the model's maximum token length to 5000. All other hyperparameters and training configurations follow those of LLaVA-OV-7B.

## C IMPLEMENTATION DETAILS FOR INFERENCE

In this section, we specifically introduce how our model works during inference to supplement the outline in Section 3.3 in the main paper. There are two key design points to meet our expectations: (1) The pipeline should work robustly for open-world applications with broad generalization abilities. (2) The pipeline should have the ability to filter and rank noisy relations. To satisfy these two requirements, SPOT disentangles relation pruning and proposal from direct relation prediction, leaving the VLM module only the task for spatial reasoning. The relation proposals are provided by out-of-the box detection models, leveraging the most recent advancements in this area. We observed that using this simple but effective approach, our framework could detect fine-grained relations and could be seamlessly applied to different scenarios, providing more comprehensive graphs compared to directly using VLMs to process everything. However, this framework raises three challenges:

- The quantity of detected bounding boxes is generally too large.

- The relation proposal sequences are overly long, making it inefficient for VLM to output long texts.

- There are many spatially implausible object pairs existing.

To mitigate these problems, we apply some strategies to improve the overall performance

**Over-detection Suppression.** In detection results, multiple bounding boxes may correspond to the same real-world object but have different yet reasonable category labels. For instance, a person may be detected as both "man" and "person," and both exist in the vocabulary. To mitigate this, we apply cross-category Non-Maximum Suppression (NMS) apart from the standard NMS to reduce overlapping boxes across semantically similar labels. For each remaining object, we aggregate all plausible labels into a label group. During evaluation, if the ground-truth label is present in this group, the object is considered correctly classified. For the real application, any label works for usage.

**Filter and Rank Relations.** To address this problem, we incorporate the classification probabilities from the object detector as object-level confidence scores, selecting the first 100 objects of higher probabilities. For relation-level filtering, we compute the distance between the centers of two objects. We explored three different types of distance for reference.

1. Standard normalized geometry distance: $d = \frac{||c_i - c_j||_2}{\sqrt{H^2 + W^2}}$,

2. Standard normlizaed 3D spatial distance: $d = \frac{||c_i^{3D} - c_j^{3D}||_2}{\sqrt{H^2 + W^2 + Z^2}}$,

3. Distance considering the object size: $d = \sqrt{d_{geo} d_{size}}$, where $d_{geo} = \frac{||c_i - c_j||_2}{\sqrt{H^2 + W^2}}$ and $d_{size} = \frac{||c_i - c_j||_2}{||c_i - c_j||_2 + (d_i + d_j)/2}$,

where $c_i$ and $c_j$ are the 2D center points; $c_i^{3D}$ and $c_j^{3D}$ are the 3D center points; $H$, $W$ and $Z$ denote the height, width and depth of the image; $d_i$ and $d_j$ are diagonal of object bounding boxes. We observe that the third distance works better than the others, and [1], [2]'s results are similar. The score for a candidate pair (i, j) is defined as: $s_{ij} = p_i \times d_{ij} \times p_j$, where $p_i$ and $p_j$ are the classification probabilities, and $d_{ij}$ is the normalized center distance. We discard relation pairs with $s_{ij} > 0.8$, as they are likely too far apart to form meaningful relations.

# D  IMPLEMENTATION DETAILS FOR BENCHMARK EVALUATION

## D.1  OVERVIEW OF EVALUATION SETTING

We follow the evaluation protocol established in Xu et al. (2017), adopting two commonly used settings: Predicate Classification (**PredCLS**) and Scene Graph Detection (**SGDet**). In PredCLS, both ground-truth object categories and bounding boxes are provided. The model receives candidate object pairs and predicts their relationships. In SGDet, the model must operate on detected objects rather than ground-truth annotations. This setting is more reflective of real-world deployment, as it introduces potential detection errors that affect downstream relation prediction. The primary distinction between PredCLS and SGDet lies in the source of object detections—ground-truth versus predicted. A key requirement in these settings is the ability to rank and filter predicted relation triplets, prioritizing the most informative and plausible ones. We evaluate this using Recall@k, where k denotes the number of top-ranked predictions considered. Additionally, to assess the overall relation prediction accuracy, we compare the model's full predicted scene graph against the ground-truth graph. In our main paper, we report both PredCLS and SGDet results for closed-set evaluation on VG150. For cross-domain evaluation and 3D whole-scene evaluation, we focus on PredCLS, where object annotations are available to isolate relation prediction performance. The evaluation setting remains consistent inside every table.

## D.2  BASELINE SELECTION AND ADAPTATION

**In-Domain Evaluation.** We evaluate a range of baseline models, which can be broadly categorized into VLM-based and non-VLM-based approaches. For non-VLM-based models, we compare against methods trained on the same dataset to ensure a fair comparison. These models typically rely on task-specific architectures and do not incorporate large pretrained vision-language models. For VLM-based baselines, we compare with another recent work PGSG (Li et al., 2024b), which relies on BLIP as backbone and uses free-form prompt: "Generate the scene graph" in the paper.

**Cross-Domain Evaluation.** We compare SPOT with other open-vocabulary approaches on new domain data which has never been seen during the training to explore and compare the open-world generalization abilities. For the VLM baselines like LLaVA-OV-7B and GPT-4o, we use the SPOT structured prompt with the addition of an in-context example output (shown in Fig. 6). For ConceptGraphs, we use their free-form prompt template (Fig. 7), with the removal of the original prompt's fixed set of relation options in order to adapt to the wider set of relations present in 3DSSG (see Fig. 6). The visual context is provided by overlaying bounding boxes on the image to indicate object locations, ensuring consistent visual grounding across models. For the other previous approaches like OvSGTR, we directly apply them to the new-source inputs with their built-in end-to-end detection and ranking modules.

```
{
"from"   "human",
"value"   "<image>\nYou are an agent specializing in identifying the physical and
spatial relationships in images for 3D mapping. \nYour task is to analyze the
images and output a list of tuples describing the physical relationships between
objects.\nNote that you are describing the **physical relationships** between the
**objects inside** the image.\nYou will also be given a text list of relation
tuples. The list will be in the format: [(\"object 1\", \"object 2\"), (\"object
3\", \"object 2\")].\nPlease predict all relations in the tuples.\nHere is the list
of predicate tuples: [(bench 0, grass 6), (sky 4, grass 6),..., (wall 8, building
7)]. Please describe the spatial relationships between the objects in all tuples.
\nAn illustrative example of the expected response format might look like this:
[("object 1", "on top of", "object 2"), ("object 3", "under", "object 2")].\nOnly
output the relation triplet list without additional explanation and symbols."
}
{
"from"   "gpt",
"value"   ""
}
```

Figure 6: Prompt template similar to SPOT for baseline evaluation

```
{
"from"   "human",
"value"   "<image>\nYou are an agent specializing in identifying the physical and
spatial relationships in images for 3D mapping. \nYour task is to analyze the
images and output a list of tuples describing the physical relationships between
objects.\nNote that you are describing the **physical relationships** between the
**objects inside** the image.\nYou will also be given a text list of the numeric
ids of the objects in the image. The list will be in the format: [\"name1 1\",
\"name2 2\", ...].\nAn illustrative example of the expected response format might
look like this: [(\"object 1\", \"on top of\", \"object 2\"), (\"object 3\",
\"under\", \"object 2\")].\nHere is the list of labels for the annotations of the
objects in the image: ['bench 0', 'bench 1', 'tree 2', 'fence 3', 'sky 4',
'pavement 5', 'grass 6', 'building 7', 'wall 8']. Please describe the spatial
relationships between the objects in the image. \nOnly output the relation triplet
list without additional explanation and symbols."
}
{
"from"   "gpt",
"value"   ""
}
```

Figure 7: Prompt template in free form, similar to ConceptGraphs for baseline evaluation

### D.3 RELATION EVALUATION

**In-Domain Evaluation.** Since the model is directly trained on the same dataset, the results can be evaluated without any vocabulary gap as in prior works. We use recall to measure how many ground truth relations are correctly predicted, without accounting for inverse or symmetric variants. Specifically, when the ground truth includes only one directional relation (e.g., "A under B" without the inverse "B under A"), the results are evaluated strictly based on the directional relation that is present in the annotations. If the model predicts the inverse (e.g., "B under A"), it is not considered correct unless it exactly matches the ground truth. While this setting ensures consistency with the prior work and fair comparison, we acknowledge that extending the evaluation protocol to account for equivalent relationships in the opposite direction is an interesting future work to explore.

**Cross-Domain Evaluation.** A common limitation in existing evaluation protocols for cross-domain datasets is that semantically similar but lexically different relations are treated as incorrect. For instance, synonymous expressions like "beside" and "next to" may convey the same meaning but only one may appear in the ground truth, leading to unfair penalization. To address this, we introduce a more semantically-aware evaluation method by leveraging a large language model (Qwen-7B) as a judge. Given a relation prediction and its corresponding ground-truth triplet, we prompt the LLM with a predefined template (shown in Figure 8) to determine whether the predicted relation is semantically equivalent to the ground truth. If the model answers yes, the prediction is considered correct. This

```
Prediction: [<mouse3> <next to> <keyboard 4>]
Ground Truth: [<mouse3> <beside> <keyboard 4>]

Prompt: Given (mouse next to keyboard), (mouse beside keyboard),
whether these two phrases indicate similar spatial relations between
two objects inside the the tuple. Only return Yes or No without
further explanation.

Answer: Yes
```

Figure 8: LLM as a judge to evaluate the similarity between two relation words

enhanced evaluation metric is reported as "Recall w/ Similarity" or "Accuracy w/ Similarity". This evaluation protocol aims to give model credit for correct relations expressed in an open vocabulary that are semantically equivalent to the ground truth.

The whole cross-domain evaluation pipeline is as follows, including some specific rules for simple cases:

- **Case 1:** prediction relation == ground truth relation → correct
- **Case 2:** prediction contains the same spatial word as the label but with the inclusion of "on", "in", or "from" → correct:
  - In our experiments on the 3DSSG dataset, we use "standing on/on", "built in/in", "hanging on/hanging from".
  - While these rules account for common cases that are clearly correct, it is difficult to iterate all plausible rules comprehensively. Thus, for all other cases, we propose to use LLM as a judge in order to assess the accuracy of the model's open vocabulary predictions against a closed vocabulary label space.
- **Case 3:** otherwise, we use LLM (Qwen-7B) to judge. The prompt is as Fig. 8:
  - We checked the results of LLM's judgment to verify its ability to judge semantically equivalent spatial phrases and found it to be accurate in practice. Given its strong performance and flexibility, we adopt the LLM-based judgement for all cross-domain tasks.

# E 3D SCENE GRAPH GENERATION PIPELINE

## E.1 PIPELINE OVERVIEW

To enable scalable and generalizable 3D-SGG in open-world settings, we follow the pipeline introduced in ConceptGraphs Gu et al. (2024), leveraging large-scale 2D foundation models for both object discovery and relation reasoning. The overall process consists of two core stages: (1) 3D object construction, which defines the graph nodes, and (2) relation prediction, which defines the graph edges.

Formally, we aim to construct a 3D scene graph $G = (V, E)$, where $V = o_1, o_2, ..., o_N$ is the set of 3D objects in the scene, and $E = (o_i, r_{ij}, o_j)$ represents the directed relations $r_{ij}$ between object pairs. Rather than training a dedicated open-vocabulary 3D segmentation model on point clouds, we use 2D segmentation Kirillov et al. (2023) and detection models to extract object information from images. Specifically, a segmentation model is used to produce instance masks, and an open-vocabulary detector provides object labels. Given a sequence of RGB-D frames along with camera intrinsic and poses, we project 2D masked pixels into 3D via depth lifting. Points from multiple views are fused into unified object-level point clouds based on geometry similarity $\phi_{geo}(i, j) = nnratio(p_{t,i}, p_{o_j})$, which is the proportion of points in point cloud $p_{t,i}$ of object $t$ in frame $i$ that have nearest neighbors in point cloud $p_{o_j}$).

For relation prediction, we treat each object pair $(o_i, o_j) \in V \times V$ as a candidate edge and use our proposed SPOT to infer the relation. The predicted relation is then assigned back to the corresponding

3D object pair using object IDs and masks. After iterating through all available frames and pruning redundant or spurious edges, we obtain the final 3D scene graph that represents the semantic and spatial structure of the environment.

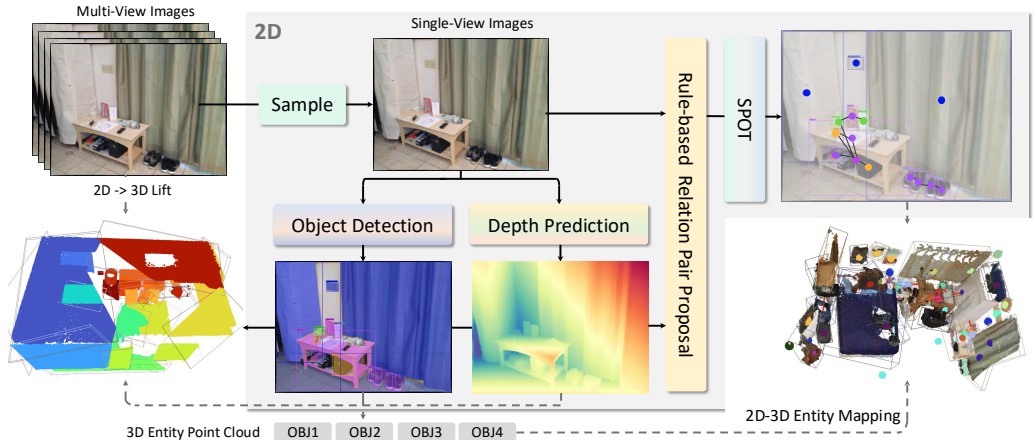

Figure 9: **Overview of the 2D-to-3D scene graph generation pipeline.** Given multi-view images, single-view frames are sampled for object detection and depth prediction. The predicted depth maps and object boxes are used to lift pixels into 3D entity point clouds. Concurrently, the 2D object boxes are utilized to generate rule-based object pair proposals, which are then passed to the SPOT to predict visual relations. Finally, the predicted 2D relations are mapped back into 3D using the pre-computed entity correspondence, resulting in a grounded 3D scene graph.

### E.2 DUPLICATED NODES REMOVAL

The pipeline is capable of combining parts of objects into a unified representation by leveraging both spatial and semantic similarity across views; however, we acknowledge that in certain cases—particularly for large, planar structures like floors or walls—this fusion can be imperfect due to limitations in frame sampling or camera motion.

The framework leverages two types of similarity scores to match newly detected objects with previously stored ones, spatially and semantically:

- **Spatial IoU:** Computed based on the overlap between lifted partial point clouds. If the IoU $> 0.5$, we consider these two objects to be spatially aligned.
- **Semantic Similarity:** A contrastive score computed from CLIP embeddings extracted from object labels, defined as:

$$\Phi_{\text{sem}}(i,j) = \frac{f_i^T f_j}{2} + \frac{1}{2} ,$$

where $f_i$ and $f_j$ are the CLIP embeddings for object $i$ and $j$. If the score $> 0.5$, we believe they belong to the same object category.

These two scores are jointly used to fuse identical objects observed from multiple views and to remove duplicates. In our experiments, we observe that small objects' point clouds can typically be fused and recovered reliably from a few frames. For large objects (e.g., floors, walls, ceilings), the fusion is sensitive to two factors:

- **Frame sampling frequency:** Sparse sampling may increase disparity between views, reducing overlap and similarity.
- **Camera motion:** Adjacent frames may have limited field-of-view overlap.

One feasible method to mitigate this issue is to increase the sampling rate to increase the adjacent overlap. However, when camera movement is too large, this problem still exists, which is a limitation

of most existing methods that follow this frame-wise approach by lifting 2D detections to 3D. This is an exciting direction for future exploration.

### E.3 EDGE CASES IN 3DSG

Since the 3D scene graph is generated from images of the scene and relies on generating 2D scene graphs as the "intermediate" step, there is a limited corner case when two objects have never been co-observed in any images, but have some semantic relation between them. This limitation is common to many current 3D scene graph generation (Gu et al., 2024) approaches that rely on 2D, per-frame observations. Like prior approaches, our method assumes that meaningful object relations emerge from co-visible object pairs within at least one view. Handling disjoint objects from sparse views remains an open challenge.

While our pipeline theoretically could predict relations for such disjoint object pairs if global relation proposals are provided, as mentioned in the question, these predictions would rely more heavily on textual priors instead of visual cues. We see this scenario as a valuable direction for future work: developing a multi-view context model allowing global relation queries across frames to support relation inference in sparse-view settings.

## F MORE QUANTITATIVE RESULTS

In this section, we provide more quantitative comparisons under cross-domain settings under PVSG benchmark. PVSG is a recent popular video benchmark. We conduct the experiment under the per-frame PrecCLS setting ($\sim$ 20k frames). SPOT is compared with open-source baselines in Table 13. As shown in the table, SPOT outperforms all baselines, further demonstrating its adaptiveness and superiority on fundamental relation prediction. What's more, we believe that a better frame-wise graph generalization ability could provide a strong potential foundation for further temporal evaluation. We would explore this integration into temporal video reasoning as future work.

Table 13: **Comparison on PVSG per-frame benchmark**, +ST indicates using our proposed structured prompt

| SGG model | PVSG per-frame dataset (PredCLS) | | | |
|---|---|---|---|---|
| | Acc. | Acc w/ sim | R @ 50 / 100 | R w/ sim @ 50 / 100 |
| *LLaVA-OV-7B (Li et al., 2024a)* | - | - | 0.76 / 0.76 | 4.41 / 4.41 |
| *Qwen2.5-VL-7B (Bai et al., 2025)* | - | - | 2.62 / 2.62 | 9.49 / 9.49 |
| *InternVL3-8B (Wang et al., 2025)* | - | - | 14.6 / 14.6 | 21.1 / 21.1 |
| *InternVL3-8B+ST (Wang et al., 2025)* | 35.5 | 51.3 | 29.0 / 32.3 | 46.5 / 49.9 |
| **SPOT** | **45.1** | **60.4** | **37.9 / 40.4** | **53.9 / 57.2** |

## G MORE QUALITATIVE RESULTS

In this section, we provide more qualitative results on different datasets. In Fig. 10, Fig. 11, Fig. 12, we separately display more visualizations on three cross-domain datasets: PSG, 3DSSG, and ScanNet Dai et al. (2017), showing the generalization ability of our model. In Fig. 13, we also present results on PSG that illustrate how SPOT predicts richer spatial relations (e.g. "behind"), even though the ground-truth answers are different. This finding raises the need for a more robust and comprehensive evaluation method to quantify different perspectives of the scene graph prediction performance.

## H LIMITATIONS

While SPOT demonstrates strong generalization by leveraging a vision-language model (VLM), it remains inherently constrained by the prior knowledge and reasoning abilities of the underlying language model. This reliance limits its ability to handle relationships or concepts that fall too differently from the pretrained distribution. Also, although we adopt the flexible and efficient pruning mechanism, it may cause the framework to ignore long-range plausible relations with a stricter threshold. In terms of efficiency, SPOT trades off throughput for generalization. Unlike traditional

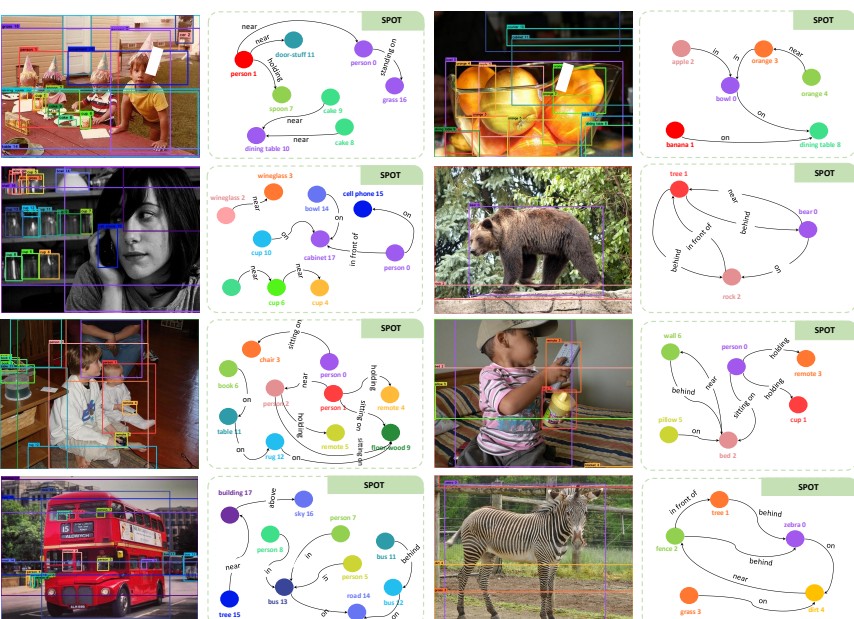

Figure 10: Additional cross-domain qualitative results on Panoptic Scene Graph Dataset. We visualize only the relations that exist in the ground truth.

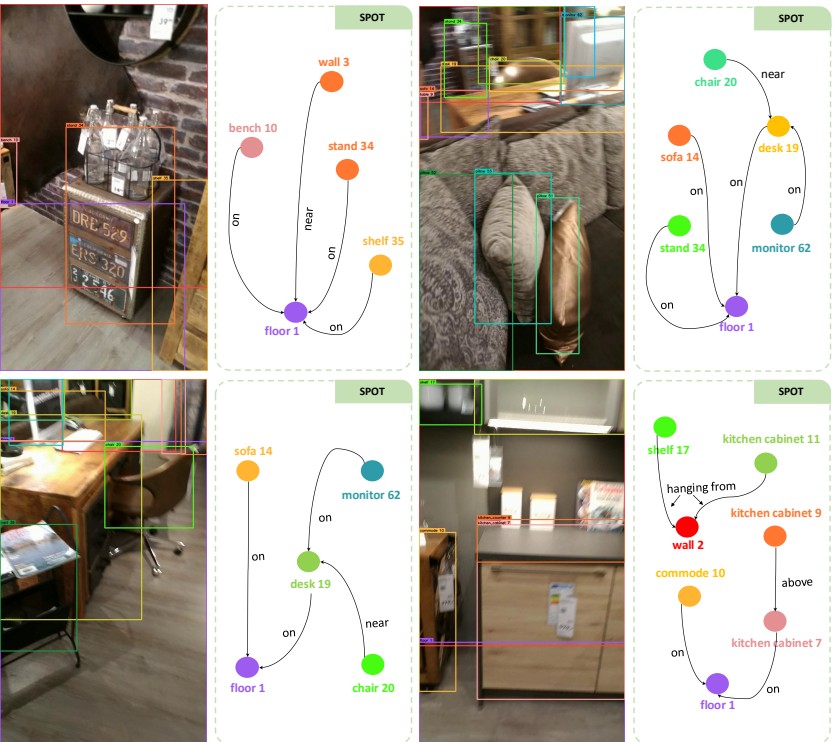

Figure 11: Additional cross-domain qualitative results on 3DSSG. We visualize only the relations that exist in the ground truth.

approaches such as OvSGTR (Chen et al., 2024), which utilize lightweight architectures like GroundingDINO and simple MLP classifiers, our VLM-based method incurs additional computational overhead. Conventional methods can predict thousands of relation pairs within seconds, whereas prompting a large language model for the same quantity of relations introduces latency and resource demands. Another potential limitation arises from the length of the prediction sequence. As the

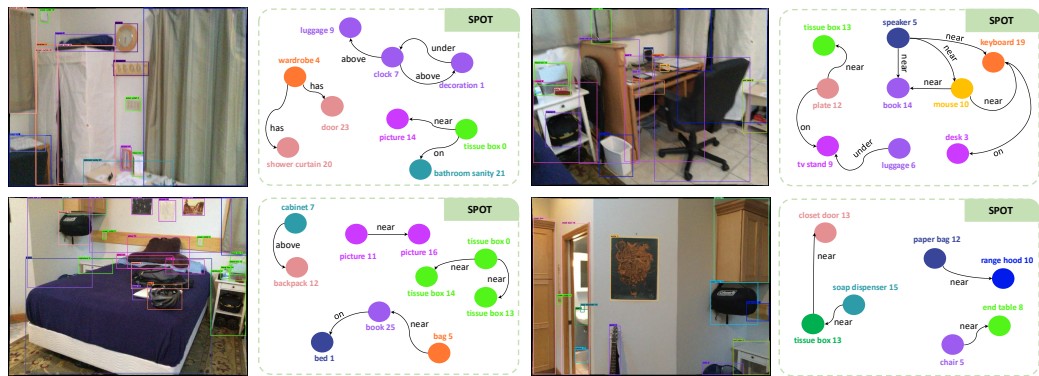

Figure 12: Qualitative cross-domain results on ScanNet. In contrast to PSG and 3DSSG, ScanNet is not a standard scene graph dataset. We include these results to show the generalization ability of SPOT and effectiveness in practical use with a real object detections.

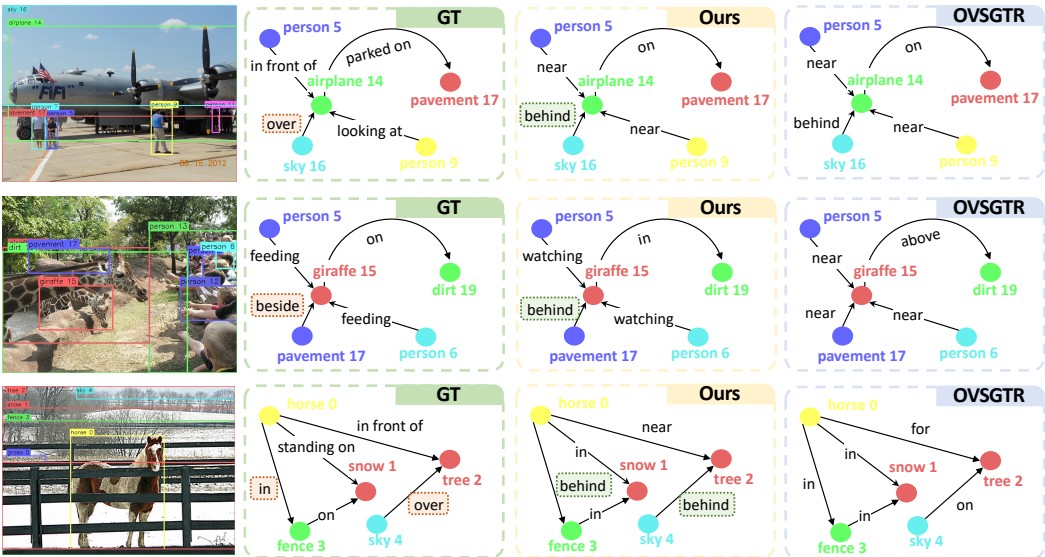

Figure 13: Additional cross-domain qualitative results on Panoptic Scene Graph Dataset. We observe that SPOT predicts richer spatial relations, which we highlight with green boxes.

number of predicted triplets grows, the prompt length increases accordingly, which can lead to degraded performance due to the model's limited context window. Although we attempt to mitigate this by segmenting long prompts into smaller batches during inference, this issue of forgetting remains a broader challenge in both the scene graph generation downstream task and general language modeling.

# I    SOCIAL IMPACT & SAFEGUARDS

A key application of 3D scene graph models is their integration into embodied agent systems for planning and interaction, where agents rely on spatial relation triplets and object locations to make decisions. Prediction of scene graphs for this purpose offers the potential for more explainable decision-making in these critical systems. However, current quantitative evaluation protocols for open-world 3D scene graph generation remain limited. To explore the full value of these systems to improve the safety of robotic applications, further quantitative vetting becomes essential to ensure that both relationship predictions and object detections faithfully represent the physical environment. Without such safeguards, erroneous predictions could result in agents' interaction with objects in unsafe or unintended ways, posing potential risks in real-world deployment.

