# OpenReview forum: "SPOT: Structured Prompting with Object-centric Tokens for open-world scene graphs"
_ICLR.cc/2026/Conference — Submitted to ICLR 2026_

### Official Review · Reviewer_8o4S · 2025-10-28

**Soundness:** 3
**Presentation:** 3
**Contribution:** 2
**Rating:** 6
**Confidence:** 3

**Summary:**

The paper introduces SPOT (Structured Prompting with Object-centric Tokens), a novel, fully open-source framework for Open-World Scene Graph Generation (SGG) that leverages Vision-Language Models (VLMs). The core problem addressed is that existing VLM/LLM-based SGG methods often rely on proprietary models (like GPT-4o), use unstructured prompts, and struggle with weak spatial reasoning, leading to vague or physically implausible relation predictions.

**Strengths:**

Strong Spatial Grounding: The core innovation of using Object-Centric Visual Features (embedding coordinate-aware visual tokens for each object) successfully counters the VLM's strong textual prior, leading to more visually grounded and accurate predictions, especially for non-canonical spatial relations.

Efficiency and Practicality: The combination of Spatially Aware Pruning and the Structured Prompt drastically improves the efficiency and output quality. Pruning reduces unnecessary VLM calls, while the prompt constrains the output, leading to higher recall and better overall scene graph quality.

Novel Evaluation Metric: The Accuracy/Recall with Similarity metric provides a necessary, robust solution for fairly evaluating open-vocabulary predictions against closed-set ground truth, a valuable contribution to the open-world SGG community.

**Weaknesses:**

Dependency on External Detection: The pipeline is fundamentally dependent on the performance of the external open-vocabulary object detector (e.g., Grounding DINO). Errors in object bounding boxes or labels produced by the detector directly limit the potential accuracy of the subsequent relation prediction module.

Heuristic Pruning: The use of a simple, rule-based spatial filter with a fixed distance threshold (0.8) is efficient but heuristic. This may result in prematurely discarding long-range, but semantically relevant, relations (e.g., "person looking at TV") that the VLM might otherwise correctly infer based on semantic context.

Evaluation: When considering relationships between nearby objects, how common relational vocabulary such as "near" or "next to" are handled is also worth considering. It remains to be assessed whether the framework proposed in the paper encourages the model to output such vague relational terms.

**Questions:**

Refer to Weaknesses.

---

> ### Author Response · Authors · 2025-11-22
>
> We appreciate the reviewer for giving inspiring feedback
>
> **W1: Dependency on external detection**
>
> **A1:** Thank you for raising this point. We agree that the quality of object detections influences later relation prediction. To demonstrate the robustness of our pipeline, we have explicitly evaluated with such errors under both PredCLS and SGDet protocols to separate these effects.
>
> * **PredCLS**: use groundtruth object boxes and labels, allowing us to isolate the relation prediction module independently from the external detection noise.
> * **SGDet**: use object proposals from real detectors, where, in our case, this is the external detection module rather than an in-built component. This setting directly reflects performance under imperfect detections.
>
> As shown in Tables 2 and 3 in the paper. SPOT consistently outperforms previous approaches, many of which rely on in-built end-to-end detectors. This demonstrates that our relation prediction is more precise, and SPOT remains robust and superior even when this external detector is imperfect and introduces errors. Furthermore, using an off-the-shelf detector allows the framework to be highly effective in the open-world setting, supporting broader category coverage and enabling efficient finetuning of the scene graph. And our plug-and-play design allows us to flexibly replace with newer updated detection models for better performance without additional training.
>
> ---
>
> **W2: Heuristic pruning**
>
> **A2:** We appreciate the reviewer for this insightful comment. This distance-based pruning represents a trade-off between coverage and efficiency. It is proposed as an effective and user-controllable heuristic that offers both efficiency and flexibility. Users could choose how expansive the relations they hope the framework to predict, hence, the threshold doesn’t hardcode the notion of long-range. For example,
>
> * Threshold \= 1.0:
>   * Disable the pruning, yielding the complete graph prediction with all long-range relation pairs included.
> * Smaller threshold:
>   * Restrict the predictions to more spatially plausible ones, shorter-range.
>
> We acknowledge that VLMs have better semantic reasoning in this case compared to heuristic pruning, but empirically, it’s not stable enough. In practice, the VLMs generally failed to output complete or guaranteed long-range graphs when prompted without constraints, even though some relations are semantically plausible. We also experimented with using a VLM (Qwen2-VL-7B in our case) to infer pruning decisions and found its behavior close to **random guessing**.  As shown in Table 9 of the manuscript (attached below), we have also tried to train the VLM to automatically prune relations (first row), which also underperforms our filtering (second row).
>
> |  | VG (In-domain 1k) | PSG (Cross-domain 1k) |  |
> | :---- | :---- | :---- | :---- |
> |  | R @ 50/100↑ | R @ 50/100↑ | R w/ Sim @ 50/100↑ |
> | VLM trained to prune | 33.0 / 42.7 | 13.2 / 15.8 | 32.1 / 38.3 |
> | Our heuristic prune | **58.2 / 63.4** | **15.9 / 18.0** | **38.7 / 44.2** |
>
> Overall, this pruning provides an effective and robust solution under this trade-off. But definitely, the usage of VLMs is an interesting and promising direction to explore in the future.
>
> ---
>
> **W3: Vague relational terms in evaluation**
>
> **A3:** Thank you to the reviewer for making this interesting observation. The point you bring up is actually a deficiency of the open-world evaluation protocol. In open-world evaluation, more broadly applicable relationships (i.e., near / next to) will naturally dominate the output predictions because they are often technically correct, and without the closed-world setting, there is no reason not to predict them. The key is to also predict additional semantically interesting relationships when relevant, even for “nearby” objects. For SPOT evaluated on PSG open-world we do find that “near/next to” are the top predicting relationship, but we also find that in general the top 10 predicted relationships (after near/next to) are: “in front of”, “attached to”, “on”, “looking at”, “over”, “enclosing”, “talking to”, “standing on”, “driving on”, “sitting on”, demonstrating that the model also has significant predictions of relationships which are more specific.

---

### Official Review · Reviewer_8srQ · 2025-10-30

**Soundness:** 2
**Presentation:** 3
**Contribution:** 1
**Rating:** 4
**Confidence:** 4

**Summary:**

This paper proposes SPOT (Structured Prompting with Object-centric Tokens), an open-source framework that adapts open-vision language models (VLMs) like LLaVA-OV for open-world scene graph generation (SGG). Here are three main contributions from SPOT:
- A structured fill-in-the-blank prompting template that explicitly constrains the model to predict relation predicates for candidate object
- Object-centric visual tokens, computed by pooling SigLIP patch embeddings inside bounding boxes and concatenating them with object text tokens.
- A distance-based relation pruning mechanism that filters implausible object pairs based on geometric proximity.

SPOT is evaluated on both 2D benchmarks (Visual Genome, Panoptic Scene Graph) and 3D datasets (3DSSG, ScanNet). Experiments show improved recall and semantic alignment compared to prior open-source models and even proprietary systems like GPT-4o, particularly in cross-domain generalization.

While the results are solid, the contribution is mainly an engineering consolidation of existing ideas (structured prompting, visual token pooling, heuristic pruning) into an open, reproducible pipeline. The “3D extension” is largely a projection-based adaptation of 2D reasoning rather than true 3D spatial modeling.

**Strengths:**

- This paper introduces a clear and reproducible system design, leveraging open-source models (LLaVA-OV + SigLIP) instead of proprietary GPT-4o.
- This paper conducts carefully experiments including: (1) comprehensive ablations: language vs. vision priors, coordinate vs. visual tokens, depth integration, pruning strategies. (2) Strong empirical validation across in-domain and cross-domain datasets, including adaptation to 3DSSG. (3) Structured prompting framework demonstrably improves recall and semantic precision across diverse VLMs (Table 1).
- This paper's writing is very clear.

**Weaknesses:**

- Limited novelty: The main contributions (templated prompting, patch pooling, distance pruning) are refinements of known techniques rather than new formulations.
- 3D reasoning overstated: with current model design, it is confusion for me to see why the method would work for multiview images. Therefore, I check out the supplementary material, which I find a lot more modules are getting introduced and I am not sure how your model gets panoptic labels from multiview images. As you stated in the paper: *Specifically, a segmentation model is used to produce instance masks, and an open-vocabulary detector provides object labels*. Is this segmentation model 2D or 3D, how to ensure instance id is bounded to a specific object? In the other hands, concept fusion is feedforward. I do not think this distinction has been mentioned in the main text.
3. Lack of comparisions of state of art models like Qwen2.5-VL/GPT-5 and more.

**Questions:**

1. This paper will be benefits if the target is 3D scenes. If SPOT only distill VG data for training, it is unclear for me to see why the 3D spatial awareness is better.
2. I see authors praises on modular models. Can the author also provide a more end to end model for the whole-scene setting?
3. Have the authors tested robustness to noisy detections or missing objects?

---

> ### Author Response · Authors · 2025-11-22
>
> We appreciate the reviewer for giving detailed feedback.
>
> **W1: Limited Novelty**
>
> **A1:** Thank you for raising this point. We would like to clarify that the novelty of SPOT lies not in a single technique, but in the effective design of the modular framework, with which one can leverage open-source VLMs with minimal training to achieve competitive or superior performance in scene graph generation compared to even proprietary models. The gains we observed are from the combination of our components, rather than one individual component.
>
> Specifically, instead of the simple free-form prompt, we adopt the structured prompt, which allows constrained but more controlled predictions, which improves coverage and suppresses hallucination, especially for small models. Our object-centric implicit feature embedding injects positional and visual cues within the prompt, which differs from prior approaches that rely on visual marks on the input image or explicit coordinate text under the SGG literature. The pruning strategy is intentionally simple but performs effectively to ensure a broad relation coverage while keeping inference efficient. The modularity of SPOT leveraging foundation models allows efficient finetuning of the scene-graph pipeline under limited data / compute settings. Together, the whole framework achieves improvements across benchmarks. We believe this integrated design constitutes the primary novelty of our work.
>
> ---
>
> **W2: 3D reasoning overstated**
>
> **A2:** Thank you for this comment. We would like to take the opportunity to clarify our approach to 3D reasoning. But first, we note that our approach is primarily designed as a mechanism to leverage strong open-source available 2D models. We additionally demonstrate that using simple techniques together with depth enables us to leverage this 2D pipeline to evaluate using our approach in 3D scene graph generation benchmarks. 3D evaluation, however, is not the primary protocol for this work, and as such, the key details describing our efforts to lift to 3D are left to the appendix. To summarize here, we take the following steps in 3D (please see Figure 9 in the appendix).
>
> 1. **2D segmentation/detection:** Run 2D object detection (GroundingDINO) / segmentation (EfficientSAM) on independent frames from the video. See Appendix E.1 for details
> 2. **Instance ID tracking:** We perform instance ID tracking across sequential frames using the following approach.
>    1. Lift 2D masks into 3D using ground truth depth maps.
>    2. Associate instance IDs: we compute a similarity score between detected objects in the current frame and the detected objects tracked from frames until now. We consider the location of said object from the prior frame and compute **Spatial IoU** and **Semantic Similarity** across object names (See Appendix E.2).
>    3. If a match is found, the observations are fused for partial point cloud, predicted relations, etc; Otherwise, a new instance ID is created.
>
> Finally, note that we do not leverage ConceptFusion in this work \- we would appreciate if the reviewer could clarify this part of the question so we can respond.

---

> ### Author Response · Authors · 2025-11-22
>
> **W3: Lack of comparison with Qwen2.5-VL / GPT-5**
>
> **A3:** We thank the reviewer for suggesting these comparisons. Note that the manuscript already includes results from Qwen2.5-VL at the time of submission. And the comparison with GPT-5 is infeasible due to the cost of large-scale dataset evaluation. To demonstrate the adaptiveness of our framework, we additionally perform new comparisons with very recent models to demonstrate the robustness of our approach.
>
> * **Qwen2.5-VL**:
>   * We reported the results at the time of submission on Qwen2.5-VL-7B in Table 1 with the free-form template and our proposed template, where an improvement is witnessed after using the structured prompt. We attached Qwen2.5-VL lines here for reference.
>
> |  | VG (1k) |  | PSG (1k) |  |
> | :---- | :---- | :---- | :---- | :---- |
> |  | R @ 50/100↑ | R w/ Sim @ 50/100↑ | R @ 50/100↑ | R w/ Sim @ 50/100↑ |
> | Qwen2.5-VL-7B (free-form) | 1.40 / 1.40 | 7.88 / 7.88 | 1.17 / 1.17 | 6.97 / 6.97 |
> | Qwen2.5-VL-7B (structured) | **3.71 / 4.23** | **18.6 / 21.1** | **3.37 / 3.71** | **21.6 / 24.0** |
>
> * **GPT-5**:
>   * **Cost**: SGG is a dense task for token input. Rerunning this dense query pattern over the full evaluation set incurs massive token usage. Higher API cost of GPT-5 makes it infeasible for us to use it for evaluation. This also demonstrates the accessibility and necessity of our proposed framework to conduct such dense relation queries.
>
> * **Other state-of-the-art models**:
>   * To better demonstrate our pipeline’s wide applicability. We tested our Structured Prompt on the **very latest models**: InternVL3.5-8B (Aug.) and Qwen3-VL-8B (Sep.). As shown in the following table, our structured prompt delivers consistent performance gains over standard free-form prompting.
>
> |  | VG (1k) |  | PSG (1k) |  |
> | :---- | :---- | :---- | :---- | :---- |
> |  | R @ 50/100↑ | R w/ Sim @ 50/100↑ | R @ 50/100↑ | R w/ Sim @ 50/100↑ |
> | InternVL3.5-8B (free-form) | 3.76 / 3.76 | 11.2 / 11.2 | 9.13 / 9.13 | 18.8 / 18.8 |
> | InternVL3.5-8B (structured) | **4.94 / 5.45** | **21.4 / 23.7** | **14.1 / 15.0** | **30.0 / 32.2** |
> | Qwen3-VL-8B (free-form) | 8.81 / 8.81 | 18.2 / 18.2 | 10.1 / 10.1 | 26.3 / 26.3 |
> | Qwen3-VL-8B (structured) | **16.1 / 18.7** | **37.1 / 42.6** | **17.4 / 19.2** | **40.3 / 44.7** |
>
> ---
>
> **Q1: Why 3D spatial awareness better**
>
> **A1:** Thanks for this question. In our framework, 3D information is only leveraged to be able to take a 2D object relation prediction and lift it properly into 3D space in order to provide a valid 3D scene graph for evaluation. Therefore, in this first version, we would **not** expect especially stronger performance due to the 3D information. Our strong performance in 3D is due to the same reason as SPOT has strong performance in 2D; we demonstrate a low-cost algorithm for leveraging the best state-of-the-art models in 2D for detection/segmentation and relation prediction. With minimal finetuning, we are able to provide a pipeline that **outperforms custom solutions, even on the 3D benchmark**. (see Table 5, main paper). These numbers could likely be further improved by more sophisticated integrations of 3D information, but we leave that effort to future work.

---

> ### Author Response · Authors · 2025-11-22
>
> **Q2: Provide a more end-to-end model for the whole scene setting**
>
> **A2:** SPOT leverages multiple off-the-shelf strong pre-trained models. This is shown to be highly effective, outperforming multiple specialized end-to-end trained systems. We argue that modularity of SPOT is a strength as it allows for fine-tuning of scene graph prediction even under lower resource (less data or compute) settings. In general, however, some of the modules leveraged, for example image encoder (SigLip in our case) or a detector, could be jointly fine-tuned if one had access to sufficient data and compute. This would then provide more of an end-to-end approach.
>
> ---
>
> **Q3: Test robustness to noisy detections or missing objects**
>
> **A3:** Yes, we explicitly tested robustness to detection noise and missing objects through the SGDet evaluation setting, reported in Table 3\.  To quantify robustness, we compare two distinct settings:
>
> * **PredCLS (ideal):** the model uses ground-truth object categories and bounding boxes, representing an oracle scenario with zero noise or missing objects.
> * **SGDet (real world / noisy):** the model uses object proposals provided from a real detector (GroundingDINO). The setting introduces real-world noise, which includes errors, noisy detections, jittered bounding boxes, and missing objects.
>
> As shown in Tables 2 & 3, in both settings, even when affected by these noisy detections, SPOT demonstrates an overall better performance compared to previous approaches, especially VLM-based methods. For example, the values in the following reference table. This demonstrates that even with noise in detection, our framework remains robust and effective even when the detection is imperfect. We paste part of the table for reference.
>
> |  | PredCLS |  |  | SGDet |  |
> | :---- | ----- | ----- | ----- | ----- | :---- |
> |  | R @ 50 | R @ 100 | Acc. | R@ 50/100 | mR @ 50 / 100 |
> | OvSGTR (No-VLM) | **60.5** | 61.9 | 60.8 | 36.4 / 42.4 | 7.20 / 8.80 |
> | PGSG (VLM) | 33.8 | 40.2 | 49.2 | 20.3 / 23.6 | **10.5** / 12.7 |
> | SPOT | 58.2 | **63.4** | **70.3** | **38.1 / 44.8** | 10.3 / **12.7** |

---

### Official Review · Reviewer_cT6c · 2025-11-01

**Soundness:** 2
**Presentation:** 3
**Contribution:** 2
**Rating:** 4
**Confidence:** 5

**Summary:**

Authors propose SPOT, an open-source novel structured prompting framework leveraging vision language models for scene graph generation. It demonstrates effective spatial and semantic reasoning, outperforming proprietary baselines like GPT-4o on open-world benchmarks.

**Strengths:**

- Authors introduce Structured Prompting with Object-centric Tokens (SPOT) for open-world scene graph generation. It combines pretrained VLMs with new techniques to enhance their relational reasoning capabilities.

- Authors design a template-based structured prompt (in contrast to the free-form prompts of prior works that more precisely guides the model to produce comprehensive scene graphs both out-of-the-box and after refinement through fine tuning.

- Authors encourage the model to consider the visual scene layout through integration of an object-centric visual feature when predicting relations. This additional signal improves upon the VLM’s standard processing of the image, increasing spatial alignment and relation accuracy on the open Visual Genome and PSG benchmarks.

- To enable open-world prediction with an external object detector and no pre-defined vocabulary, authors propose leveraging spatially aware pruning and integrating flexibility during evaluation to minimize penalties for semantically similar predictions. This approach goes beyond the protocol which exhaustively constructs a fully-connected graph over all object pairs, which proves computationally expensive and suffers from redundant and irrelevant relation predictions.

**Weaknesses:**

1) The authors chose GroundingDINO as their open-vocabulary object detector, but the paper does not provide a quantitative justification for why they chose this method.

2) The comparison with SOTA graph generation methods is incomplete; various efficient graph generation methods like FROSS [1] exist.

[1] Hou, H. Y., Lee, C. Y., Sonogashira, M., & Kawanishi, Y. (2025). FROSS: Faster-than-Real-Time Online 3D Semantic Scene Graph Generation from RGB-D Images. In Proceedings of the IEEE/CVF International Conference on Computer Vision (pp. 28818–28827).

3) Methods for encoding individual objects in images are known and have been successfully applied in Chat-Scene [1], GPT4Scene [2], and others.

[1] Haifeng Huang, Yilun Chen, Zehan Wang, Rongjie Huang, Runsen Xu, Tai Wang, Luping Liu, Xize Cheng, Yang Zhao, Jiangmiao Pang, et al. Chat-scene: Bridging 3D scenes and large language models with object identifiers. In The Thirty-eighth Annual Conference on Neural Information Processing Systems, 2024

[2] Zhangyang Qi, Zhixiong Zhang, Ye Fang, Jiaqi Wang, and Hengshuang Zhao. Gpt4scene: Understand 3D scenes from videos with vision-language models. arXiv preprint arXiv:2501.01428, 2025.

4) The authors state that SPOT is a scalable and fully open-source framework; however, no links to the anonymous repository could be found in the appendices or in the text of the article to confirm this statement.

5) The article's formatting should be improved; punctuation marks are missing at the end of the formulas.

**Questions:**

1) The authors write that SPOT is a scalable and fully open-source framework, but no links to an anonymous repository could be found in the appendices or the text of the article to confirm this claim. How will the method perform on other popular benchmarks that evaluate the quality of graph generation, such as STAR [1] and PVSG [2]?

[1] Wu, B., Yu, S., Chen, Z., Tenenbaum, J. B., & Gan, C. (2024). Star: A benchmark for situated reasoning in real-world videos. arXiv preprint arXiv:2405.09711.

[2] Yang, J., Peng, W., Li, X., Guo, Z., Chen, L., Li, B., ... & Liu, Z. (2023). Panoptic video scene graph generation. In Proceedings of the IEEE/CVF Conference on Computer Vision and Pattern Recognition (pp. 18675–18685).

---

> ### Author Response · Authors · 2025-11-22
>
> We appreciate the reviewer for the suggestive feedback.
>
> **W1: Why choose GroundingDINO**
>
> **A1:** We thank the reviewer for raising this point. We selected GroundingDINO as it is known to be a strong detector with benchmarked performance out of the box in open-world settings. We experimented internally with multiple detection models and found GroundingDINO to work for our purposes. However, we appreciate the reviewer's request to verify quantitatively that this is indeed the best backbone model for our method. We therefore provide a quantitative comparison below against four other recent state-of-the-art detection models. Indeed, we confirm that GroundingDINO outperforms the alternative methods.
>
> Specifically, GroundingDINO is compared against other popular detection tools, including VLM-based model (Qwen3-VL-8B), Vision foundation model (Florence-2-large), and classic real-time detector (YOLO-series).
>
> * **Quantitative Comparison (Recall vs Efficiency):** We evaluate these models on a small subset of the Visual Genome dataset in an **open-vocabulary** setting without any parameter searching. For models requiring text inputs, the whole dataset vocabulary space is provided.
>   * **VLM-based approach:** While VLMs like Qwen3-VL-8B are flexible and work well when the input vocab is rather small, they are slow (47s/image) for pipeline integration and struggle with increasingly more out-of-context errors when the object space grows large, which is our case.
>   * **Classic approach:** While YOLOv11 is extremely fast, its recall is lower than GroundingDINO, which would bottleneck the following relation prediction performance. Furthermore, it is closed-set and does not support flexible text input.
>
> |  | Object Recall % | Speed per image (s) |
> | :---- | :---- | :---- |
> | Qwen3-VL-8B  | 3.48 | 47.0 |
> | Florence-2-Large | 13.6 | 0.472 |
> | YOLOv11n | 10.2 | 0.0451 |
> | YOLOv11x | 10.7 | 0.0542 |
> | GroundingDINO | 20.9 | 0.261 |
>
>
> ---
>
> **W2: Lack comparison with efficient graph generation methods, such as FROSS\[U\]**
>
> **A2:** Thank you for bringing this recent work to our attention. FROSS is not directly applicable to our setting; the two methods target different scopes: FROSS is evaluated under the 3D **closed-set** setting using its custom design for efficiency, whereas our 3D experiments for SPOT  leverage our 2D pipeline by lifting to 3D to evaluate under an **open-vocab** setting. Additionally, FROSS (ICCV’25) was first released on arXiv on July 26, 2025, and updated later on Aug 10\. Under ICLR’s conference guidelines, this paper qualifies as **concurrent** work.
>
> ---
>
> **W3: Encoding individual objects has been applied in previous works**
>
> **A3:** We appreciate the reviewer for highlighting these interesting works. While GPT4Scene and Chat-Scene leverage the encoding of individual objects, their underlying intuition and approach differ from SPOT.  Specifically, GPT4Scene adds objects’ cues by marking visual labels on BEV images as an image input. The object-level inputs are applied strictly at the level of the input image as visual annotations, whereas SPOT embeds object-centric features at the latent token level within the structured prompt. Chat-Scene proposes to leverage an external 2D / 3D encoder to aggregate the object features from text, point cloud, and multi-view images for object-level QA/grounding tasks. SPOT instead embeds implicit positional and visual information into our dense structured prompts through efficient visual feature reuse to enable more accurate open-world relation prediction. Also, the application scenarios differ: GPT4Scene and Chat-Scene focus on general 3D QA and grounding, whereas SPOT targets the open-vocabulary scene graph task.
>
> Moreover, our contribution extends beyond object encoding, incorporating a structured prompt and a relation proposal pipeline to augment open-source VLMs with dense relation reasoning using minimal training.
>
> We thank the reviewers for this suggestion and for pointing out these related works. We include them in the updated reference to make our literature more complete.

---

> ### Author Response · Authors · 2025-11-22
>
> **W4: No links to an anonymous repository**
>
> **A4:** We thank the reviewer for highlighting the importance of code availability. We strongly believe in open-source research and are firmly committed to releasing our code and model checkpoints upon acceptance. We would like to clarify that one of our key contribution claims is that our method builds on top of open-source components. So not only will our model and code be publicly released, but others can run our method and reproduce our results without requiring access to proprietary large-scale models.
>
> To further ensure reproducibility and open-source readiness, we have included extensive implementation details in the manuscript.
>
> * **Open-weight base model:**
>   * Our framework is built entirely on the open-source LLaVA-OV-7B backbone and utilizes public benchmarks (VG, PSG, 3DSSG).
> * **Implementation details:**
>   * We have provided comprehensive implementation details, especially in the Supplementary Materials, to allow for replication. Specifically, Appendix B elaborates on data preprocessing, structured prompt, hyperparameters during training; Appendix C details the inference pipeline and relation pruning; Appendix D describes the evaluation protocol and pseudo-code.
>
> ---
>
> **W5: Formatting should be improved**
>
> **A5:** Thank you for pointing this out. We update the manuscript to include the missing punctuation marks at the end of the equations. The corrections are highlighted in magenta.
>
> ---
>
> **Q1: Performance on other benchmarks**
>
> **A1:** Thank you for suggesting these popular benchmarks. We note that both PVSG and STAR are designed for video-level SGG, where temporal reasoning, object tracking, and cross-frame relation propagation are essential. Since SPOT is a frame-wise general 2D SGG framework, it cannot be directly applied to full video sequences without additional temporal modules. Therefore, we report per-frame comparisons with the baselines in our paper and leave the video-level evaluation for future work.
>
> * **PVSG per-frame comparison**:
>   * SPOT is evaluated on the PVSG validation set under the per-frame PredCLS setting (\~20k frames). We compare it with the open-source baselines included in our paper. Due to the large scale of the evaluation frames, we use InternVL3-8B with our proposed structured prompt template as a substitute for proprietary models such as GPT-4o with template, which is not feasible due to the cost.
>   * As shown in the table, SPOT outperforms all baselines, further demonstrating its adaptiveness and superiority across benchmarks.
>
> | Table: PVSG per-frame dataset (PredCLS) |  |  |  |  |
> | :---- | :---- | :---- | :---- | :---- |
> | Method | Acc | Acc w/ sim | R @ 50 / 100 | R w/ sim @ 50 / 100 |
> | LLaVA-OV-7B (free-form prompt) | \- | \- | 0.76 / 0.76 | 4.41 / 4.41 |
> | Qwen2.5-VL-7B (free-form prompt) | \- | \- | 2.62 / 2.62 | 9.49 / 9.49 |
> | InternVL3-8B (free-form prompt) | \- | \- | 14.6 / 14.6 | 21.1 / 21.1 |
> | InternVL3-8B (structured prompt) | 35.5 | 51.3 | 29.9 / 32.3 | 46.5 / 49.9 |
> | SPOT (Ours) | **45.1** | **60.4** | **37.9 / 40.4** | **53.9 / 57.2** |
>
> * **PVSG temporal metrics and STAR benchmark**:
>   * The temporal SGG evaluation over the whole sequence in PVSG and the STAR benchmark relies heavily on temporal reasoning, object tracking, and dynamic relations prediction across time. Extending SPOT to the video domain and further downstream tasks would require additional temporal and symbolic reasoning modules, which are non-trivial and beyond the scope of our current proposed 2D open-world framework.
>   * However, we believe that a better frame-wise graph generation ability could provide a strong potential foundation for these tasks. We are excited to explore the integration of SPOT into temporal video reasoning and downstream tasks as future work.

---

### Official Review · Reviewer_oinB · 2025-11-04

**Soundness:** 3
**Presentation:** 3
**Contribution:** 3
**Rating:** 6
**Confidence:** 3

**Summary:**

This work introduces Structured Prompting with Object-centric Tokens (SPOT) for scene graph generation in open world scenarios. SPOT shows strong performance on 2D benchmarks and can be further extended to 3D scene graph generation.

**Strengths:**

1. This work introduced SPOT to augment VLMs’ ability on open-world scene graph generation
2. The experiment results and ablation study results are extensive and good.

**Weaknesses:**

1. In the proposed model, is there any design difference for processing spatial relation and semantic relation? What is the evaluation for each category of relations in In-Domain Evaluation?
2. In Cross-Domain Evaluation, what is the generalization performance of the proposed model in spatial relation and semantic relation?

**Questions:**

Please refer to the Weaknesses section.

---

> ### Author Response · Authors · 2025-11-22
>
> We appreciate the reviewer for giving useful feedback
>
> **W1.1: Design differences for spatial & semantic relation**
>
> **A1.1:** In our proposed framework, we don’t introduce any separate architectural components for spatial and semantic subsets. Both types of predicates are handled uniformly through the structured prompting and backbone. We clarify that the framework shows better spatial and semantic reasoning is based on observed improvement naturally from our design:
>
> * For spatial relationships: benefit from the implicit positional embedding for object-level feature input and distilled knowledge from data.
> * For semantics relationships: benefit from the richer relational patterns present in the training data and underlying knowledge of VLMs (especially between person-person, person-object cases)
>
> We see improvements across both types of relations from the emergent properties of structured prompt training with object-centric visual features. Additionally, some relations may have both spatial and semantic properties (e.g., “walking on”). SPOT’s prompting framework allows the model to combine both spatial and semantic information for predicting relationships.
>
> ---
>
> **W1.2: Evaluation for in-domain evaluation**
>
> **A1.2:** For in-domain evaluation, since the model is trained directly on the same dataset, there is no vocabulary gap to compare with previous works; hence, we take the strict evaluation protocol for both the semantic and spatial relations, which we also detail in Appendix D.3 In-domain Evaluation. The pseudocode is:
>
> * Case1: predicted relation \== ground truth relation → return correct
> * Case2: otherwise → incorrect
>
> This also follows the standard evaluation protocol to have a fair comparison with the previous approaches for close-vocabulary evaluation.
>
> ---
>
> **W2: Generalization performance in cross-domain evaluation**
>
> **A2:** Thank you for this insightful question. Our manuscript reports the overall cross-domain generalization performance in Tables 4,5, where SPOT consistently outperforms prior works in both recall and accuracy. We appreciate the reviewer’s suggestion to further analyze the spatial and semantic components separately. To explore this, we conducted an additional quantitative evaluation on PSG by manually grouping relations into “spatial” and “semantic” subsets. However, we note that some relations have both spatial and semantic properties in the open-world setting. For example, “walking on” is both a spatial (“on”) and semantic (“walking”) relationship and so cannot be cleanly separated into one of the two types. Nevertheless, we report the split results below for reference, placing any relation with semantic characteristics into the semantic category:
>
> |  | Spatial |  | Semantics |  |
> | :---- | ----- | :---- | ----- | :---- |
> |  | Acc | Acc w/ sim | Acc | Acc w/ sim |
> | InternVL-3-8B | 22.5 | 44.9 | 11.8 | 27.9 |
> | Qwen2.5-VL-7B | 6.81 | 47.7 | 7.33 | 19.8 |
> | OvSGTR | 10.2 | 44.6 | 0.45 | 39.8 |
> | **SPOT (Ours)** | **39.3** | **69.1** | **12.5** | **50.2** |
>
> Across both subsets, SPOT demonstrates clear improvements compared to all baselines, contributing to an overall better performance.

---

### Author Response · Authors · 2025-12-02
**Rebuttal Summary**

Dear Area Chair,

We would like to thank you and all reviewers for the time and effort in evaluating our submission. To aid your decision-making, we summarize key points from our rebuttal.

1. **Key contribution:** Our key contribution is the design of a new modular framework, SPOT, for open-world scene graph generation. By leveraging open-source VLMs with minimal retraining, SPOT achieves superior performance compared to state-of-the-art methods while requiring only modest model sizes, limited training data, and practical compute budgets.

2. **Comparison to proprietary VLMs:** Using our framework, open-source VLMs with minimal finetuning are able to outperform proprietary models (i.e., GPT-4o).

3. **Choice of detectors:** We justify our choice of GroundingDINO through added quantitative comparison against classic and VLM-based detectors, showing it offers the best accuracy-speed trade-off (cT6c-W1). Additionally, we demonstrate its robustness to noisy external detectors using SGDet results, where SPOT consistently outperforms prior approaches (8srQ-Q3, 8o4S-W1).

4. **Other rebuttal experiments summarized:** The rebuttal adds empirical analysis on cross-domain generalization, showing superior performance on both fine-grained spatial and semantic subsets (oinB-W2). Additional per-frame PVSG benchmark results (cT6c-Q1) and comparisons with the most recent VLMs (8srQ-W3) are included, where SPOT consistently delivers gains.

5. **Further clarification:** We further clarify that 3D experiments lift 2D predictions rather than proposing a custom 3D approach, to scope our contribution accurately and to address the 3D reasoning questions (8srQ-W2,Q1), and we highlight SPOT’s modularity versus end-to-end systems, while still allowing joint finetuning if resources permit (8srQ-Q2).  Finally, we specify the architecture design (oinB-W1.1), in-domain evaluation (oinB-W1.2), and heuristic pruning (8o4S-W2), discuss limitations of vague relational terms in open-world evaluation (8o4S-W3), and reaffirm our commitment to releasing code and checkpoints for open-sourcing (cT6c-W4).

Revisions have been highlighted in **magenta** in the updated manuscript.

Thank you again for your time and consideration.

Best regards,

Authors of Paper 4145

---

### Meta-Review · Area_Chair_gnea · 2026-01-10

**Summary:**

The paper introduces SPOT, a framework for open-world scene graph generation (SGG) that replaces proprietary models with an open-source pipeline. Reviewers appreciate the structured prompting and object-centric tokens which improve spatial grounding and recall. However, concerns remain regarding limited technical novelty (seen as engineering consolidation), heuristic pruning that may miss long-range relations, and a weak 3D extension that lacks true spatial modeling.

**Reviewer Concerns:**

Addressed in Rebuttal: The choice of GroundingDINO (Reviewer cT6c). Writing quality and formula formatting. Clarifications on the open-source status and repository access.

Outstanding: Novelty vs. Engineering: The core components are viewed as refinements of existing techniques (Reviewer 8srQ). Reviewer 8srQ pointed out that "The main contributions (templated prompting, patch pooling, distance pruning) are refinements of known techniques rather than new formulations." The authors focus more on explaining specifically how the method is implemented, whereas the reviewer is concerned with the technical improvements over existing methods. Regarding the "3D reasoning overstated" point mentioned by the reviewer, the authors do not explicitly answer whether it is overstated, yes or no. Reviewer 8srQ will likely not raise the score.

**Reviewer Scores:**

See Reviewer Concerns.

---

### Decision · Program_Chairs · 2026-01-26

Reject